# On Divergence Measures for Training GFlowNets

**Tiago da Silva**    **Eliezer de Souza da Silva**    **Diego Mesquita**
{tiago.henrique, eliezer.silva, diego.mesquita}@fgv.br
School of Applied Mathematics
Getulio Vargas Foundation
Rio de Janeiro, Brazil

## Abstract

Generative Flow Networks (GFlowNets) are amortized samplers of unnormalized distributions over compositional objects with applications to causal discovery, NLP, and drug design. Recently, it was shown that GFlowNets can be framed as a hierarchical variational inference (HVI) method for discrete distributions. Despite this equivalence, attempts to train GFlowNets using traditional divergence measures as learning objectives were unsuccessful. Instead, current approaches for training these models rely on minimizing the log-squared difference between a proposal (forward policy) and a target (backward policy) distribution. In this work, we first formally extend the relationship between GFlowNets and HVI to distributions on arbitrary measurable topological spaces. Then, we empirically show that the ineffectiveness of divergence-based learning of GFlowNets is due to the large gradient variance of the corresponding stochastic objectives. To address this issue, we devise a collection of provably variance-reducing control variates for gradient estimation based on the REINFORCE leave-one-out estimator. Our experimental results suggest that the resulting algorithms often accelerate training convergence when compared against previous approaches. All in all, our work contributes by narrowing the gap between GFlowNet training and HVI, paving the way for algorithmic advancements inspired by the divergence minimization viewpoint.

## 1   Introduction

The approximation of intractable distributions is one of the central issues in machine learning and modern statistics [7, 35]. In reinforcement learning (RL), a recurring goal is to find a diverse set of high-valued state–action trajectories according to a reward function. This problem may be cast as sampling trajectories proportionally to the reward, which is generally an intractable distribution over the environment [3, 9, 38, 50]. Similarly, practical Bayesian inference and probabilistic models computations involve assessing intractable posterior distributions [36, 78, 102]. In the variational inference (VI) approach, circumventing this intractability involves searching for a tractable approximation to the target distribution within a family of parametric models. Conventionally, the problem reduces to minimizing a divergence measure, such as Kullback-Leibler (KL) divergence [7, 36, 92] or Renyi-$\alpha$ divergence [51, 70], between the variational approximation and the target.

In particular, Generative Flow Networks (GFlowNets) [3, 4, 48] are a recently proposed family of variational approximations well-suited for distribution over compositional objects (e.g., graphs and texts). GFlowNets have found empirical success within various applications from causal discovery [15, 16], NLP [30], and chemical and biological modeling [3, 32]. In a nutshell, a GFlowNet learns an iterative generative process (IGP) [26] over an extension of the target's support, which, for sufficiently expressive parameterizations of transition kernels, yields independent and correctly distributed samples [3, 48]. Remarkably, training GFlowNets typically consists of minimizing the log-squared difference between a proposal and target distributions over the extended space via SGD [4, 55], contrasting with divergence-minimizing algorithms commonly used in VI [7, 72].

Indeed, Malkin et al. [56] suggests that trajectory balance (TB) loss training for GFlowNets leads to better approximations of the target distribution than directly minimizing the reverse and forward KL divergence, particularly in setups with sparser rewards. Nevertheless, as we highlight in Section 3, these results are a potential consequence of biases and high variance in gradient estimates for the divergence's estimates, which can be overlooked in the evaluation protocol reliant upon sparse target distributions. Therefore, in Section 5, we present a comprehensive empirical investigation of the minimization of well-known $f$-divergence measures (including reverse and forward KL), showing it is an effective procedure that often accelerates the training convergence of GFlowNets relative to alternatives. To achieve these results, we develop in Section 4 a collection of control variates (CVs) [63, 71] to reduce the variance without introducing bias on the estimated gradients, improving the efficiency of the optimization algorithms [77, 85]. In summary, our *main contributions* are:

1. We evaluate the performance of forward and reverse KL- [47], Renyi-$\alpha$ [74] and Tsallis-$\alpha$ [91] divergences as learning objectives for GFlowNets through an extensive empirical campaign and highlight that they frequently outperform traditionally employed loss functions.

2. We design control variates for the gradients of GFlowNets' divergence-based objectives. Therefore, it is possible to perform efficient evaluations of the optimization objectives using automatic differentiation frameworks [69], and the resulting experiments showcase the significant reduction in the variance of the corresponding estimators.

3. We developed a theoretical connection between GFlowNets and VI beyond the setup of finitely supported measures [56, 112], establishing results for arbitrary topological spaces.

## 2 Revisiting the relationship between GFlowNets and VI

Initially, we review Lahlou et al. [48]'s work on GFlowNets for distributions on topological spaces, a perspective applied consequentially to obtain the equivalence between GFlowNets training and VI divergence minimization in a more generic setting. Finally, we describe standard variance reduction techniques for solving stochastic optimization problems.

**Notations.** Let $(\mathcal{S}, \mathcal{T})$ be a topological space with topology $\mathcal{T}$ and $\Sigma$ be the corresponding Borel $\sigma$-algebra. Also, let $\nu\colon \Sigma \to \mathbb{R}_+$ be a measure over $\Sigma$ and $\kappa_f, \kappa_b\colon \mathcal{S} \times \Sigma \to \mathbb{R}_+$ be transition kernels over $\mathcal{S}$. For each $(B_1, B_2) \in \Sigma \times \Sigma$, we denote by $\nu \otimes \kappa(B_1, B_2) := \int_{B_1} \nu(\mathrm{d}s) k(s, B_2)$. Likewise, we recursively define the *product kernel* as $\kappa^{\otimes 0}(s, \cdot) = \kappa(s, \cdot)$ and, for $n \geq 1$, $\kappa^{\otimes n}(s, \cdot) = \kappa^{\otimes n-1}(s, \cdot) \otimes \kappa$ for a transition kernel $\kappa$ and $s \in \mathcal{S}$. Note, in particular, that $\kappa^{\otimes n}$ is a function from $\mathcal{S} \times \Sigma^{\otimes n+1}$ to $\mathbb{R}_+$, with $\Sigma^{\otimes n+1}$ representing the product $\sigma$-algebra of $\Sigma$ [1, 96]. Moreover, if $\mu$ is an absolutely continuous measure relatively to $\nu$, denoted $\mu \ll \nu$, we write $\mathrm{d}\mu/\mathrm{d}\nu$ for the corresponding density (Radom-Nikodym derivative) [1]. Furthermore, we denote by $\mathcal{P}(A) = \{S\colon S \subseteq A\}$ the power-set of a set $A \subset \mathcal{S}$ and by $[d] = \{1, \dots, d\}$ the first $d$ positive integers.

**GFlowNets.** A GFlowNet is, in its most general form, built upon the concept of a *measurable pointed directed acyclic graph* (DAG) [48], which we define next. Intuitively, it extends the notion of a *flow network* to arbitrary measurable topological spaces, replacing the directed graph with a transition kernel specifying how the underlying states are connected.

**Definition 1** (Measurable pointed DAG [48]). Let $(\bar{\mathcal{S}}, \mathcal{T}, \Sigma)$ be a measurable topological space endowed with a reference measure $\nu$ and forward $\kappa_f$ and backward $\kappa_b$ kernels. Also, let $s_o \in \bar{\mathcal{S}}$ and $s_f \in \bar{\mathcal{S}}$ be distinguished elements in $\bar{\mathcal{S}}$, respectively called *initial* and *final* states, and $\mathcal{S} = \bar{\mathcal{S}} \setminus \{s_f\}$. We assume $\{s_f\}$ is open. A *measurable pointed DAG* is then a tuple $(\mathcal{S}, \mathcal{T}, \Sigma, \kappa_f, \kappa_b, \nu)$ satisfying:

1. **(Terminality)** If $\kappa_f(s, \{s_f\}) > 0$, then $\kappa_f(s, \{s_f\}) = 1 \; \forall s \in \bar{\mathcal{S}}$. Also, $\kappa_f(s_f, \cdot) = \delta_{s_f}$.

2. **(Reachability)** For all $B \in \Sigma$, $\exists n \in \mathbb{N}$ s.t. $\kappa_f^{\otimes n}(s_o, B) > 0$, i.e., $B$ is reachable from $s_o$.

3. **(Consistency)** For every $(B_1, B_2) \in \Sigma \times \Sigma$ such that $(B_1, B_2) \notin \{(s_o, s_o), (s_f, s_f)\}$, $\nu \otimes \kappa_f(B_1, B_2) = \nu \otimes \kappa_b(B_2, B_1)$. Moreover, $\kappa_b(s_o, B) = 0$ for every $B \in \Sigma$.

4. **(Continuity)** $s \mapsto \kappa_f(s, B)$ is continuous for $B \in \Sigma$.

5. **(Finite absorption)** There is a $N \in \mathbb{N}$ such that $\kappa_f^{\otimes N}(s, \cdot) = \delta_{s_f}$ for every $s \in \mathcal{S}$. We designate the corresponding DAG as *finitely absorbing*.

In this setting, the elements in the set $\mathcal{X} = \{s \in \mathcal{S} \setminus \{s_f\}\colon \kappa_f(s, \{s_f\}) > 0)\}$ are called *terminal states*. Illustratively, when $\mathcal{S}$ is finite and $\nu$ is the counting measure, the preceding definition

corresponds to a connected DAG with an edge from $s \in \mathcal{S}$ to $s' \in \mathcal{S}$ iff $\kappa_f(s, \{s'\}) > 0$, with condition 5) ensuring acyclicity and condition 2) implying connectivity. A GFlowNet, then, is characterized by a measurable pointed DAG, a potentially unnormalized distribution over terminal states $\mathcal{X}$ and learnable transition kernels on $\mathcal{S}$ (Definition 2). Realistically, its goal is to find an IGP over $\mathcal{S}$ which, starting at $s_o$, samples from $\mathcal{X}$ proportionally to a given positive function.

**Definition 2** (GFlowNets [48]). A *GFlowNet* is a tuple $(\mathcal{G}, P_F, P_B, \mu)$ composed of a measurable pointed DAG $\mathcal{G}$, a $\sigma$-finite measure $\mu \ll \nu$, and $\sigma$-finite Markov kernels $P_F \ll \kappa_f$ and $P_B \ll \kappa_b$, respectively called *forward* and *backward* policies.

**Training GFlowNets.** In practice, we denote by $p_{F_\theta} : \mathcal{S} \times \mathcal{S} \to \mathbb{R}_+$ the density of $P_F$ relative to $\kappa_f$, which we parameterize using a neural network with weights $\theta$. Similarly, we denote by $p_B$ the density of $P_B$ wrt $k_b$. Our objective is, for a given *target measure* $R \ll \mu$ on $\mathcal{X}$ with $r = {}^{\mathrm{d}R}/_{\mathrm{d}\mu}$, estimate the $\theta$ for which the distribution over $\mathcal{X}$ induced by $P_F(s_o, \cdot)$ matches $R$, i.e., for every $B \in \Sigma$,

$$\sum_{n \geq 0} \int_{\mathcal{S}^n} p_{F_\theta}^{\otimes n}(s_o, s_{1:n}, s_f) \mathbb{1}_{s_n \in B} \kappa_f^{\otimes n}(s_o, \mathrm{d}s_{1:n}) = \frac{R(B)}{R(\mathcal{X})}.$$

Importantly, the above sum contains only finitely many non-zero terms due to the finite absorption property of $\kappa_f$. To ensure that $p_{F_\theta}$ abides by this equation, Lahlou et al. [48] showed it suffices that one of the next *balance conditions* are concomitantly satisfied by $P_F$ and $P_B$.

**Definition 3** (Trajectory balance condition). For all $n \geq 0$ and $\mu^{\otimes n}$-almost surely $\forall s_{1:n} \in \mathcal{S}^n$, $p_{F_\theta}^{\otimes n}(s_o, s_{1:n}, s_f) = \frac{r(s_n)}{Z_\theta} p_B^{\otimes n}(s_n, s_{n:1}, s_o)$, w/ $Z_\theta$ denoting the target distribution's partition function.

**Definition 4** (Detailed balance condition). For an auxiliary parametric function $u : \mathcal{S} \to \mathbb{R}_+$ and $\mu^{\otimes 2}$-almost surely on $(s, s') \in \mathcal{S}^2$, $u(s)p_{F_\theta}(s, s') = u(s')p_B(s', s)$ and $u(x) = r(x)$ for $x \in \mathcal{X}$.

To enforce the trajectory balance (TB) or detailed balance (DB) conditions, we conventionally define a stochastic optimization problem to minimize the expected log-squared difference between the left- and right-hand sides of the corresponding condition under a probability measure $\xi$ supported on an appropriate space [4, 15, 48, 49, 52, 55, 67]. For TB, e.g., we let $\xi$ be defined on $\Sigma^{\otimes N}$ with support $\mathrm{supp}(\mu^{\otimes N})$. Then, we estimate the GFlowNet's parameters $\theta$ by minimizing

$$\mathbb{E}_{\tau \sim \xi} \left[ \left( \sum_{0 \leq i \leq N-1} \left( \log \frac{p_{F_\theta}(s_i, s_{i+1})}{p_B(s_{i+1}, s_i)} \right) + \log \frac{Z_\theta}{r(s_h)} \right)^2 \right] \tag{1}$$

with $\tau = s_{o:N}$ and $h = \max\{i : s_i \neq s_f\}$ being the last non-final state's index in $\tau$. As denoted by the subscript on $Z_\theta$, using the TB loss incurs learning the target's normalizing constant. While tuning $p_B$ during training is also possible, the common practice is to keep it fixed.

Henceforth, we will consider the measurable space of *trajectories* $(\mathcal{P}_\mathcal{S}, \Sigma_P)$, with $\mathcal{P}_\mathcal{S} = \{(s, s_1, \ldots, s_n, s_f) \in \mathcal{S}^{n+1} \times \{s_f\} : 0 \leq n \leq N-1\}$ and $\Sigma_P$ as the $\sigma$-algebra generated by $\bigcup_{n=1}^{N+1} \Sigma^{\otimes n}$. For notational convenience, we use the same letters for representing the measures and kernels of $(\mathcal{S}, \Sigma)$ and their natural product counterparts in $(\mathcal{P}_\mathcal{S}, \Sigma_P)$, which exist by Carathéodory extension's theorem [96]; for example, $\nu(B) = \nu^{\otimes n}(B)$ for $B = (B_1, \ldots, B_n) \in \Sigma^{\otimes n}$ and $p_{F_\theta}(\tau | s_o; \theta)$ is the density of $P_F^{\otimes n+1}(s_o, \cdot)$ for $\tau = (s_o, s_1, \ldots, s_n, s_f)$ relatively to $\mu^{\otimes n}$. In this case, we will write $\tau$ for a generic element of $\mathcal{P}_\mathcal{S}$ and $x$ for its terminal state (which is unique by Definition 1). For a comprehensive overview of GFlowNets, please refer to [48, 55].

**GFlowNets and VI.** GFlowNets can be interpreted as hierarchical variational models by framing the forward policy $p_{F_\theta}(\tau | s_o; \theta)$ in $(\mathcal{P}_\mathcal{S}, \Sigma_P)$ as a proposal to $\frac{r(x)}{Z} p_B(\tau | x)$. Malkin et al. [56] demonstrated that, for discrete target distributions, the TB loss in (1) aligns with the KL divergence in terms of expected gradients. Extending this, our Proposition 1 establishes that this relationship also holds for distributions over arbitrary topological spaces.

**Proposition 1** (TB loss- and KL divergence gradients for topological spaces). *Let $\mathcal{L}_{TB}(\tau; \theta) = \left( \log {}^{Z p_{F_\theta}(\tau | s_o; \theta)}/_{r(x) p_B(\tau | x)} \right)^2$ and $p_B(\tau) = \frac{r(x)}{Z} p_B(s_{n-1:o} | x)$ for $\tau = (s_o, \ldots, s_{n-1}, x, s_f)$. Then,*

$$\nabla_\theta \mathbb{E}_{\tau \sim P_F(s_o, \cdot)} [\mathcal{L}_{TB}(\tau; \theta)] = 2\nabla_\theta \mathcal{D}_{KL}[P_F \| P_B], \tag{2}$$

*where $\mathcal{D}_{KL}[p_{F_\theta} \| p_B] = \mathbb{E}_{\tau \sim P_F(s_o, \cdot)} \left[ \log {}^{p_{F_\theta}(\tau | s_o; \theta)}/_{p_B(\tau)} \right]$ is the KL divergence between $P_F$ and $P_B$.*

This proposition shows that minimizing the on-policy TB loss is theoretically comparable to minimizing the KL divergence between $P_F$ and $P_B$ in terms of convergence speed. Since the TB loss requires estimating the intractable $R(\mathcal{X})$, the KL divergence, which avoids this estimation, can be a more suitable objective. Our experiments in Section 5 support this, with proofs provided in Appendix C. Extending this result to general topological spaces broadens the scope of divergence minimization strategies, extending guarantees for discrete spaces to continuous and mixed spaces. This generalization aligns with advances in generalized Bayesian inference [45] and generalized VI in function spaces [95], via optimization of generic divergences. We make the method theoretically firm and potentially widely applicable by proving the equivalence in these broader contexts.

**Variance reduction.** A naive Monte Carlo estimator for the gradient in Equation 2 has high variance [19], impacting the efficiency of stochastic gradient descent [97]. To mitigate this, we use *control variates*—random variables with zero expectation added to reduce the estimator's variance without bias [63, 71]. This method, detailed in Section 4, significantly reduces noise in gradient estimates and pragmatically improves training convergence, as shown in the experiments in Section 5.

## 3 Divergence measures for learning GFlowNets

This Section presents four different divergence measures for training GFlowNets and the accompanying gradient estimators for stochastic optimization. Regardless of the learning objective, recall that our goal is to estimate $\theta$ by minimizing a discrepancy measure $D$ between $P_F$ and $P_B$ that is globally minimized if and only if $P_F = P_B$, i.e.,

$$\theta^* = \arg\min_{\theta} D(P_F, P_B),\tag{3}$$

in which $P_B$ is typically fixed and $P_F$'s density $p_F^\theta$ is parameterized by $\theta$.

### 3.1 Renyi-$\alpha$ and Tsallis-$\alpha$ divergences

Renyi-$\alpha$ [74] and Tsallis-$\alpha$ [91] are families of statistical divergences including, as limiting cases, the widespread KL divergence (Section 3.2) [58]; see Definition 5. These divergences have been successfully applied to both variational inference [51] and policy search for model-based reinforcement learning [18]. Moreover, as we highlight in Section 5, their performance is competitive with, and sometimes better than, traditional learning objectives for GFlowNets based on minimizing log-squared differences between proposal and target distributions.

**Definition 5** (Renyi-$\alpha$ and Tsallis-$\alpha$ divergences)**.** Let $\alpha \in \mathbb{R}$. Also, let $p_{F_\theta}$ and $p_B$ be GFlowNet's forward and backward policies, respectively. Then, the *Renyi-$\alpha$ divergence* between $P_F$ and $P_B$ is

$$\mathcal{R}_\alpha(P_F||P_B) = \frac{1}{\alpha - 1} \log \int_{\mathcal{P}_\mathcal{S}} p_{F_\theta}(\tau|s_o)^\alpha p_B(\tau)^{1-\alpha} \kappa_f(s_o, d\tau).$$

Similarly, the *Tsallis-$\alpha$ divergence* between $P_F$ and $P_B$ is

$$\mathcal{T}_\alpha(P_F||P_B) = \frac{1}{\alpha - 1} \left( \int_{\mathcal{P}_\mathcal{S}} p_{F_\theta}(\tau|s_o)^\alpha p_B(\tau)^{1-\alpha} \kappa_f(s_o, d\tau) - 1 \right).$$

From Definition 5, we see that both Renyi-$\alpha$ and Tsallis-$\alpha$ divergences transition from a mass-covering to a mode-seeking behavior as $\alpha$ ranges from $-\infty$ to $\infty$. Regarding GFlowNet-training, this flexibility suggests that $\mathcal{R}_\alpha$ and $\mathcal{T}_\alpha$ are appropriate choices both, e.g., for carrying out Bayesian inference [16] — where interest lies in obtaining an accurate approximation to a posterior distribution —, and for combinatorial optimization [109] — where the goal is to find a few high-valued samples. Additionally, the choice of $\alpha$ provides a mechanism for controlling which trajectories are preferentially sampled during training, with larger values favoring the selection of trajectories leading to high-probability terminal states, resembling the effect of $\varepsilon$-greedy [55], thompson-sampling [73], local-search [40], and forward-looking [65, 66] techniques for carrying out off-policy training of GFlowNets [56].

To illustrate the effect of $\alpha$ on the learning dynamics of GFlowNets, we show in Figure 1 an *early stage* of training to sample from a homogeneous mixture of Gaussian distributions by minimizing Renyi-$\alpha$ divergence for different values of $\alpha$; see Section 5.1 for details on this experiment. At this stage, we note that the GFlowNet covers the target distribution's modes but fails to separate them when $\alpha$ is large and negative. In contrast, a large positive $\alpha$ causes the model to focus on a single

high-probability region. Therefore, the use of an intermediate value for $\alpha = 0.5$ culminates in a model that accurately approximates the target distribution. Also, our early experiments suggested the persistence of such dependence on $\alpha$ for diverse learning tasks, with $\alpha = 0.5$ leading to the best results. Thus, we fix $\alpha = 0.5$ throughout our experimental campaign.

Importantly, we need only the gradients of $\mathcal{R}_\alpha$ and $\mathcal{T}_\alpha$ for solving the optimization problem in Equation 3 and, in particular, learning the target distribution's normalizing constant is unnecessary, as we underline in the lemma below. This property distinguishes such divergence measures from both TB and DB losses in Equation 1 and, in principle, simplifies the training of GFlowNets.

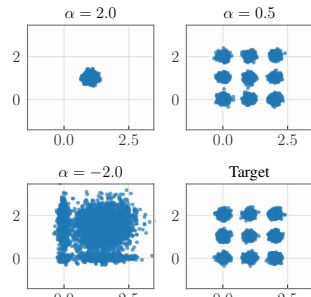

**Lemma 1** (Gradients for $\mathcal{R}_\alpha$ and $\mathcal{T}_\alpha$). *Let $\theta$ be the parameters of $p_{F_\theta}$ in Definition 5 and, for $\tau \in \mathcal{P}_\mathcal{S}$, $g(\tau, \theta) = \left(p_B(\tau|x)r(x)/p_{F_\theta}(\tau|s_o;\theta)\right)^{1-\alpha}$. The gradient of $\mathcal{R}_\alpha$ wrt $\theta$ is*

$$\nabla_\theta \mathcal{R}_\alpha(p_{F_\theta}||p_B) = \frac{\mathbb{E}[\nabla_\theta g(\tau,\theta) + g(\tau,\theta)\nabla_\theta \log p_{F_\theta}(\tau|s_o;\theta)]}{(\alpha-1)\mathbb{E}[g(\tau,\theta)]};$$

*the expectations are computed under $P_F$. Analogously, the gradient of $\mathcal{T}_\alpha$ wrt $\theta$ is*

Figure 1: Mode-seeking ($\alpha = 2$) versus mass-covering ($\alpha = -2$) behaviour in $\alpha$-divergences.

$$\nabla_\theta \mathcal{T}_\alpha(p_{F_\theta}||p_B) \stackrel{C}{=} \frac{\mathbb{E}[\nabla_\theta g(\tau,\theta) + g(\tau,\theta)\nabla_\theta \log p_{F_\theta}(\tau|s_o;\theta)]}{(\alpha-1)},$$

*in which $\stackrel{C}{=}$ denotes equality up to a multiplicative constant.*

Lemma 1 uses the REINFORCE method [97] to compute the gradients of both $\mathcal{R}_\alpha$ and $\mathcal{T}_\alpha$, and we implement Monte Carlo estimators to approximate the ensuing expectations based on a batch of trajectories $\{\tau_1, \ldots, \tau_N\}$ sampled during training [56]. Also, note that the function $g$ is computed outside the log domain and, therefore, particular care is required to avoid issues such as numerical underflow of the unnormalized distribution [3, 87]. In our implementation, we sample an initial batch of trajectories $\{\tau_i\}_{i=1}^N$ and compute the maximum of $r$ among the sampled terminal states in log space, i.e., $\log \tilde{r} = \max_i \log r(x_i)$. Then, we consider $\log \tilde{r}(x) = \log r(x) - \log \tilde{r}$ as the target's unnormalized log density. In Section 4, we will consider the design of variance reduction techniques to decrease the noise level of gradient estimates and possibly speed up the learning process.

### 3.2 Kullback-Leibler divergence

The KL divergence [47] is a limiting member of the Renyi-$\alpha$ and Tsallis-$\alpha$ families of divergences, derived when $\alpha \to 1$ [70], and is the most widely deployed divergence measure in statistics and machine learning. To conduct variational inference, one regularly considers both the *forward* and *reverse* KL divergences, which we review in the definition below.

**Definition 6** (Forward and reverse KL). *The forward KL divergence between a target $P_B$ and a proposal $P_F$ is $\mathcal{D}_{KL}[P_B||P_F] = \mathbb{E}_{\tau \sim P_B(s_f, \cdot)} \left[\log p_B(\tau)/p_{F_\theta}(\tau|s_o)\right]$. Also, the reverse KL divergence is defined by $\mathcal{D}_{KL}[P_F||P_B] = \mathbb{E}_{\tau \sim P_F(s_o, \cdot)} \left[\log p_{F_\theta}(\tau|s_o)/p_B(\tau)\right]$.*

Remarkably, we cannot use a simple Monte Carlo estimator to approximate the forward KL due to the presumed intractability of $P_B$ (which depends directly on $R$). As a first approximation, we could estimate $\mathcal{D}_{KL}[P_B||P_F]$ via importance sampling w/ $P_F$ as a proposal distribution as in [56]:

$$\mathcal{D}_{KL}[P_B||P_F] = \mathbb{E}_{\tau \sim P_F}\left[\frac{p_B(\tau)}{p_{F_\theta}(\tau|s_o)} \log \frac{p_B(\tau)}{p_{F_\theta}(\tau|s_o)}\right], \tag{4}$$

and subsequently implement a REINFORCE estimator to compute $\nabla_\theta \mathcal{D}_{KL}[P_B||P_F]$. Nevertheless, as we only need the divergence's derivatives to perform SGD, we apply the importance weights directly to the gradient estimator. We summarize this approach in the lemma below.

**Lemma 2** (Gradients for the KL divergence). *Let $\theta$ be the parameters of $P_F$ and $s(\tau;\theta) = \log p_{F_\theta}(\tau|s_o;\theta)$. Then, the gradient of $\mathcal{D}_{KL}[P_F||P_B]$ relatively to $\theta$ satisfies*

$$\nabla_\theta \mathcal{D}_{KL}[P_F||P_B] = \mathbb{E}_{\tau \sim P_F(s_o, \cdot)}\left[\nabla_\theta s(\tau;\theta) + \log \frac{p_{F_\theta}(\tau|s_o)}{p_B(\tau|x)r(x)}\nabla_\theta s(\tau;\theta)\right]$$

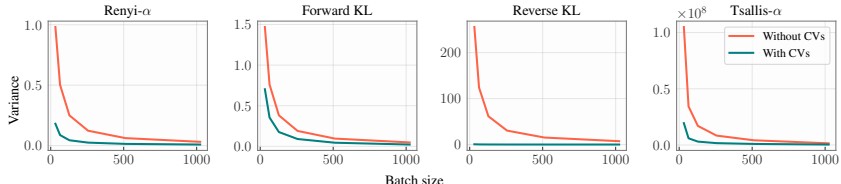

Figure 2: **Variance of the estimated gradients as a function of the trajectories' batch size.** Our control variates greatly reduce the estimator's variance, even for relatively small batch sizes.

*Correspondingly, the gradient of $\mathcal{D}_{KL}[P_B||P_F]$ wrt $\theta$ is*

$$\nabla_\theta \mathcal{D}_{KL}[P_B||P_F] \overset{C}{=} -\mathbb{E}_{\tau \sim P_F(s_o, \cdot)} \left[ \frac{p_{F_\theta}(\tau|s_o)}{p_B(\tau|x)r(x)} \nabla_\theta s(\tau; \theta) \right].$$

Crucially, choosing an appropriate learning objective is an empirical question that one should consider on a problem-by-problem basis — similar to the problem of selecting among Markov chain simulation techniques [27]. In particular, a one-size-fits-all solution does not exist; see Section 5 for a thorough experimental investigation. Independently of the chosen method, however, the Monte Carlo estimators for the quantities outlined in Lemma 2 are of potentially high variance and may require a relatively large number of trajectories to yield a reliable estimate of the gradients [97]. The following sections demonstrate that variance reduction techniques alleviate this issue.

## 4 Control variates for low-variance gradient estimation

**Control variates.** We first review the concept of a control variate. Let $f: \mathcal{P}_{\mathcal{S}} \to \mathbb{R}$ be a real-valued measurable function and assume that our goal is to estimate $\mathbb{E}_{\tau \sim \pi}[f(\tau)]$ according to a probability measure $\pi$ on $\Sigma_P$ (see Section 2 to recall the definitions). The variance of a naive Monte Carlo estimator for this quantity is $\mathrm{Var}_\pi(f(\tau))/n$. On the other hand, consider a random variable (RV) $g: \mathcal{P}_{\mathcal{S}} \to \mathbb{R}$ positively correlated with $f$ and with known expectation $\mathbb{E}_\pi[g(\tau)]$. Then, the variance of a naive Monte Carlo for $\mathbb{E}_\pi[f(\tau) - a(g(\tau) - \mathbb{E}_\pi[g(\tau)])]$ for a *baseline* $a \in \mathbb{R}$ is

$$\frac{1}{n} \left[ \mathrm{Var}_\pi(f(\tau)) - 2a\mathrm{Cov}_\pi(f(\tau), g(\tau)) + a^2 \mathrm{Var}_\pi(g(\tau)) \right], \tag{5}$$

which is potentially smaller than $\frac{1}{n}\mathrm{Var}_\pi(f(\tau))$ if the covariance between $f$ and $g$ is sufficiently large. Under these conditions, we choose the value of $a$ that minimizes Equation 5 [94], namely, $a = \mathrm{Cov}_\pi(f(\tau),g(\tau))/\mathrm{Var}_\pi(g(\tau))$. We then call the function $g$ a *control variate* [63]. Also, although the quantities defining the best baseline $a$ are generally unavailable in closed form, one commonly uses a batch-based estimate of $\mathrm{Cov}_\pi(f(\tau), g(\tau))$ and $\mathrm{Var}_\pi(g(\tau))$; the incurred bias is generally negligible relatively to the reduced variance [71, 79, 85]. For vector-valued RVs, we let $a$ be a diagonal matrix and exhibit, in the next proposition, the optimal baseline minimizing the covariance matrix's trace.

**Proposition 2** (Control variate for gradients). *Let $f, g: \mathcal{P}_{\mathcal{S}} \to \mathbb{R}^d$ be vector-valued functions and $\pi$ be a probability measure on $\mathcal{P}_{\mathcal{S}}$. Consider a* baseline *$a \in \mathbb{R}$ and assume $\mathbb{E}_\pi[g(\tau)] = 0$. Then,*

$$\underset{a \in \mathbb{R}}{\arg\min} \, \mathrm{Tr} \, \mathrm{Cov}_\pi[f(\tau) - a \cdot g(\tau)] = \frac{\mathbb{E}_\pi[g(\tau)^T(f(\tau) - \mathbb{E}_\pi[f(\tau')])]}{\mathbb{E}_\pi[g(\tau)^T g(\tau)]}.$$

Note that, when implementing the REINFORCE gradient estimator, the expectation we wish to estimate may be generally written as $\mathbb{E}_{P_F(s_o, \cdot)}[\nabla_\theta f(\tau) + f(\tau)\nabla_\theta \log p_{F_\theta}(\tau)]$. For the second term, we use a leave-one-out estimator [85]; see below. For the first term, we use $\nabla_\theta \log p_{F_\theta}$ as a control variate, which satisfies $\mathbb{E}_{P_F(s_o, \cdot)}[\nabla_\theta \log p_{F_\theta}(\tau|s_o; \theta)] = 0$. Importantly, estimating the optimal baseline $a^\star$ in Proposition 2 cannot be done efficiently due to the non-linear dependence of the corresponding Monte Carlo estimator on the sample-level gradients [2]; i.e., it cannot be represented as a vector-Jacobian product, which is efficient to compute in reverse-mode automatic differentiation (*autodiff*) frameworks [8, 69]. Consequently, we consider a linear approximation of both numerator and denominator defining $a^\star$ in Proposition 2, which may be interpreted as an instantiation of the

delta method [83, Sec. 7.1.3]. Then, given a batch $\{\tau_1, \ldots, \tau_N\}$ of trajectories, we instead use

$$\hat{a} = \frac{\left\langle \sum_{n=1}^{N} \nabla_\theta \log p_{F_\theta}(\tau_n), \sum_{n=1}^{N} \nabla_\theta f(\tau_n) \right\rangle}{\epsilon + \left\| \sum_{n=1}^{N} \nabla_\theta \log p_{F_\theta}(\tau_n) \right\|^2} \tag{6}$$

as the REINFORCE batch-based estimated baseline; $\langle \cdot, \cdot \rangle$ represents the inner product between vectors. Intuitively, the numerator is roughly a linear approximation to the covariance between $\nabla_\theta \log p_{F_\theta}$ and $\nabla_\theta f$ under $P_F$. In contrast, the denominator approximately measures the variance of $\nabla_\theta \log p_{F_\theta}$, and $\epsilon > 0$ is included to enhance numerical stability. As a consequence, for the reverse KL divergence, $\nabla_\theta f(\tau) = \nabla_\theta \log p_{F_\theta}(\tau)$, $\hat{a} \approx 1$ and the term corresponding to the expectation of $\nabla_\theta f(\tau)$ vanishes. We empirically find that this approach frequently reduces the variance of the estimated gradients by a large margin (see Figure 2 above and Section 5 below).

**Leave-one-out estimator.** We now focus on obtaining a low-variance estimate of $\mathbb{E}_{\tau \sim P_F(s_o, \cdot)}[f(\tau) \nabla_\theta \log p_{F_\theta}(\tau)]$. As an alternative to the estimator of Proposition 2, Shi et al. [85] and Salimans and Knowles [82] proposed a sample-dependent baseline of the form $a(\tau_i) = \frac{1}{N-1} \sum_{1 \le n \le N, n \ne i} f(\tau_n)$ for $i \in \{1, \ldots, N\}$. The resulting estimator,

$$\delta = \frac{1}{N} \sum_{n=1}^{N} \left( f(\tau_n) - \frac{1}{N-1} \sum_{j=1, j \ne i}^{N} f(\tau_j) \right) \nabla_\theta \log p_{F_\theta}(\tau_n),$$

is unbiased for $\mathbb{E}\left[ f(\tau) \nabla_\theta \log p_{F_\theta}(\tau) \right]$ due to the independence between $\tau_i$ and $\tau_j$ for $i \ne j$. Strikingly, $\delta$ can be swiftly computed with *autodiff*: if $\mathbf{f} = (f(\tau_n))_{n=1}^{N}$ and $\mathbf{p} = (\log p_{F_\theta}(\tau_n))_{n=1}^{N}$, then

$$\delta = \nabla_\theta \frac{1}{N} \left\langle \text{sg} \left( \mathbf{f} - \frac{1}{N-1}(\mathbf{1} - \mathbf{I})\mathbf{f} \right), \mathbf{p} \right\rangle, \tag{7}$$

with sg as the stop-gradient operation (e.g., `lax.stop_gradient` in JAX [8] and `torch.detach` in PyTorch [69]). Importantly, these techniques incur a minimal computational overhead to the stochastic optimization algorithms relative to the considerable reduction in variance they enact.

**Relationship with previous works.** Importantly, Malkin et al. [56] used $\hat{a} = \frac{1}{N} \sum_n f(\tau_n)$ as baseline and an importance-weighted aggregation to adjust for the off-policy sampling of trajectories, introducing bias in the gradient estimates and relinquishing guarantees of the optimization procedure. A learnable baseline independently trained to match $\hat{b}$ was also considered. This potentially entailed the inaccurate conclusion that the TB and DB are superior to standard divergence-based objectives. Indeed, the following section underlines that such divergence measures are sound and practical learning objectives for GFlowNets for a range of tasks.

**Illustration of the control variates' effectiveness.** We train the GFlowNets using increasingly larger batches of $\{2^i : i \in [\![5, 10]\!]\}$ trajectories with and without CVs. In this setting, Figure 2 showcases the drastic reduction in the variance, represented by the covariance matrix's trace, of the estimated learning objectives' gradients w.r.t. the model's parameters promoted by the CVs. Impressively, as we show in Figure 6, this approach significantly increases the efficiency of the underlying stochastic optimization algorithm. See Section 5 and Appendix D for further details.

## 5 Training GFlowNets with divergence measures

Our experiments serve two purposes. Firstly, we show in Section 5.2 that minimizing divergence-based learning objectives leads to competitive and often better approximations than the alternatives based on log-squared violations of the flow network's balance. This underlines the effectiveness of well-established divergence measures for training GFlowNets [56, 112]. Secondly, we highlight in Section 5.3 that the reduction of variance enacted by our control variates critically accelerates the convergence of GFlowNets. We consider widely adopted benchmark tasks from GFlowNet literature, described in Section 5.1, contemplating both discrete and continuous target distributions. Please refer to Appendix B and Appendix E for additional information on the experimental setup.

### 5.1 Generative tasks

Below, we provide a high-level characterization of the generative tasks used for synthetic data generation and training. For a more rigorous description in the light of Section 2, see Appendix B.

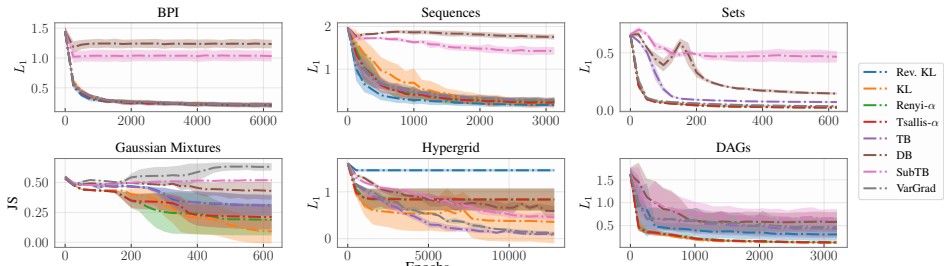

Figure 3: **Divergence-based learning objectives often lead to faster training than TB loss.** Notably, contrasting with the experiments of [56], there is no single best loss function always conducting to the fastest convergence rate, and minimizing well-known divergence measures is often on par with or better than minimizing the TB loss in terms of convergence speed. Results were averaged across three different seeds. Also, we fix $\alpha = 0.5$ for both Tsallis-$\alpha$ and Renyi-$\alpha$ divergences.

**Set generation** [3, 34, 65, 66]. A state $s$ corresponds to a set of size up to a given $S$ and the terminal states $\mathcal{X}$ are sets of size $S$; a transition corresponds to adding an element from a deposit $\mathcal{D}$ to $s$. The IGP starts at an empty set, and the log-reward of a $x \in \mathcal{X}$ is $\sum_{d \in x} f(d)$ for a fixed $f : \mathcal{D} \to \mathbb{R}$.

**Autoregressive sequence generation** [32, 55]. Similarly, a state is a seq. $s$ of max size $S$ and a terminal state is a seq. ended by an end-of-sequence token; a transition appends $d \in \mathcal{D}$ to $s$. The IGP starts with an empty sequence and, for $x \in \mathcal{X}$, $\log r(x) = \sum_{i=1\ldots|x|} g(i)f(x_i)$ for functions $f, g$.

**Bayesian phylogenetic inference (BPI)** [111]. A state $s$ is a forest composed of binary trees with labeled leaves and unlabelled internal nodes, and a transition amounts to joining the roots of two trees to a newly added node. Then, $s$ is terminal when it is a single connected tree — called a *phylogenetic tree*. Finally, given a dataset of nucleotide sequences, the reward function is the unnormalized posterior over trees induced by J&C69's mutation model [37] and a uniform prior.

**Hypergrid navigation** [3, 55, 56, 66]. A state $s \in \{0, \ldots, H-1\}^d$ is a component of a $H$-sized and $d$-dimensional Euclidean grid. The IGP starts at $\mathbf{0}$ and, if we let $\delta_i$ be the $i$-th line of the identity matrix and $\Delta(s) = \{\delta_i : i \in \{1, \ldots, d\} \wedge \max_j(s + \delta_i)_j < H\}$, a transition either adds a $\delta \in \Delta(s)$ to $s$ or stops at $s$. We use Malkin et al. [55, Section 5.1]'s reward function with $R_o = 10^{-3}$.

**Bayesian structure learning** [15, 16]. A state $s$ is a DAG representing a Bayesian network; a transition either adds an edge to $s$ or stops the IGP. Similarly to Deleu et al. [15], we ensure the added edges preserve the state's acyclicity. The reward function is defined as the maximum likelihood of the linear Gaussian structural model induced by the current state based on a fixed i.i.d. data set.

**Mixture of Gaussians (GMs)** [48, 110]. The IGP starts at $\mathbf{0} \in \mathbb{R}^d$ and proceeds by sequentially substituting each coordinate with a sample from a real-valued distribution. For a $K$-component GM, the reward of $\mathbf{x} \in \mathbb{R}^d$ is defined as $\sum_k \alpha_k \mathcal{N}(\mathbf{x}|\mu_k, \Sigma_k)$ with $\alpha_k \geq 0$ and $\sum_k \alpha_k = 1$.

**Banana-shaped distribution** [57, 76]. We use the same IGP implemented for a bi-dimensional GM. For $\mathbf{z} \in \mathbb{R}^2$, we set $r(\mathbf{x})$ to a normal likelihood defined on a quadratic function of $\mathbf{x}$, see Equation 8 in the supplement. We use HMC samples as ground truth to gauge performance on this task.

## 5.2 Assessing convergence speed

Next, we provide evidence that minimizing divergence-based objectives frequently leads to faster convergence than minimizing the standard TB loss [55].

**Experimental setup.** We compare the convergence speed in terms of the rate of decrease of a measure of distributional error when using different learning objectives for a GFlowNet trained to sample from each of the distributions described in Section 5.1. For discrete distributions, we adopt the evaluation protocols of previous works [3, 54, 55, 66] and compute the $L_1$ distance between the learned $p_T(x; \theta)$ and target $r(x)$, namely, $\sum_{x \in \mathcal{X}} |p_T(x; \theta) - r(x)/Z|$. To approximate $p_T$, we use a Monte Carlo estimate of $p_T(x; \theta) = \mathbb{E}_{\tau \sim P_B(x, \cdot)}[p_{F_\theta}(\tau|s_o; \theta)/p_B(\tau|x)]$. For continuous distributions, we echo [48, 110] and compute Jensen-Shannon's divergence between $P_T(x; \theta)$ and $R(x)$:

$$\mathcal{D}_{JS}[P_T||R] = \tfrac{1}{2}\left(\mathcal{D}_{KL}[P_T||M] + \mathcal{D}_{KL}[R||M]\right) = \mathbb{E}_{x \sim P_T}\left[\log p_T(x)/m(x)\right] + \mathbb{E}_{x \sim R}\left[\log r(x)/Zm(x)\right],$$

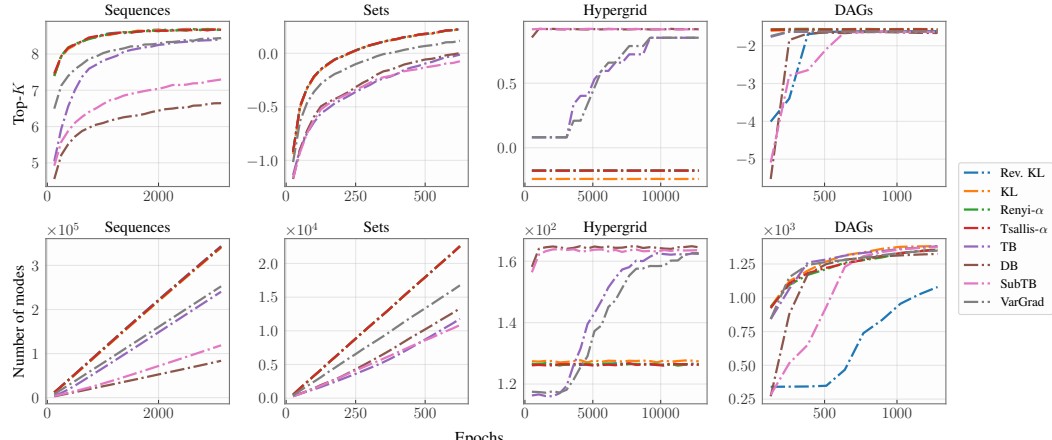

Figure 4: **Average reward for the $K$ highest scoring samples (top-K) and Number of Modes** found during training for the tasks of sequence design, set generation, hypergrid and DAG environments. With the only exception of the hypergrid task, the minimization of divergence-based measures leads to similar and often faster discovery of high-valued states relatively to their balance-based counterparts.

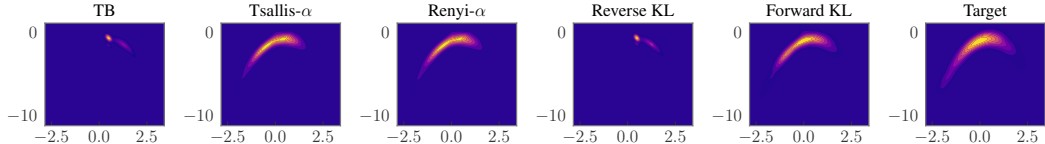

Figure 5: **Learned distributions for the banana-shaped target.** Tsallis-$\alpha$, Renyi-$\alpha$ and for. KL leads to a better model than TB and Rev. KL, which behave similarly — as predicted by Proposition 1.

with $M(B) = 1/2 \left( P_T(B) + {}^{R(B)}/_{R(\mathcal{X})} \right)$ being the averaged measure of $P_T$ and $R$ and $m$ its corresponding density relatively to the reference measure $\mu$. Remarkably, for the GMs distribution, we can directly sample from the target to estimate $\mathcal{D}_{KL}[R||M]$, and the autoregressive nature of the generative process ensures that $p_T(x) = p_{F_\theta}(\tau|s_o)$ for the unique trajectory $\tau$ starting at $s_o$ and finishing at $x$. Hence, we get an unbiased estimate of $\mathcal{D}_{KL}[P_T||M]$. Finally, let $X_t$ be the first $t$ terminal states encountered during training and $\{x_{(1)}, \ldots, x_{(k)}\}$ be an descending ordering of $X_t$ according to $r$. Then, we select a threshold $\tau \in \mathbb{R}$ and an integer $K$ to report $\mathrm{NoM}(X_t) = |\{r(x): r(x) \geq \tau \wedge x \in X_t\}|$, called *number of modes*, and $\mathrm{TopK}(X_t) = \mathrm{AVG}\left( \{r(x_{(i)}): 1 \leq i \leq K\} \right)$, referred to as *top-K average reward*. Both NoM and TopK are metrics of substantial interest in the GFlowNet literature [3, 55, 56, 64, 65].

**Results.** Figure 3 shows that the procedure minimizing divergence-based measures accelerates the training convergence of GFlowNets, whereas Figure 5 (for the banana-shaped distribution) and Table 1 highlight that we obtain a more accurate model with a fix compute budget. The difference between learning objectives is not statistically significant for the BPI task. Also, we may attribute the superior performance of reverse KL compared to the forward in the sequence generation task to the high variance of the importance-

Table 1: Divergence minimization achieves better than or similar accuracy compared to enforcing TB.

|  | BPI | Sequences | Sets | GMs |
|---|---|---|---|---|
| TB | $0.22_{\pm 0.04}$ | $0.28_{\pm 0.06}$ | $0.07_{\pm 0.00}$ | $0.31_{\pm 0.08}$ |
| Rev. KL | $0.21_{\pm 0.04}$ | $\mathbf{0.16}_{\pm 0.06}$ | $\mathbf{0.03}_{\pm 0.00}$ | $0.31_{\pm 0.09}$ |
| For. KL | $0.22_{\pm 0.04}$ | $0.23_{\pm 0.12}$ | $\mathbf{0.03}_{\pm 0.00}$ | $\mathbf{0.09}_{\pm 0.10}$ |
| Renyi-$\alpha$ | $0.22_{\pm 0.03}$ | $0.23_{\pm 0.10}$ | $\mathbf{0.03}_{\pm 0.00}$ | $0.19_{\pm 0.13}$ |
| Tsallis-$\alpha$ | $0.21_{\pm 0.04}$ | $0.22_{\pm 0.09}$ | $\mathbf{0.03}_{\pm 0.00}$ | $0.21_{\pm 0.11}$ |

sampling-based gradient estimates. Indeed, the observed differences disappear when we increase the batch of trajectories to reduce the estimator's variance (see Figure 8 in Appendix D). In conclusion, our empirical results based on experiments testing diverse generative settings and expanding prior art [48, 56, 112], shows that training methods based on minimizing $f$-divergence VI objectives with adequate CVs implemented are practical and effective in many tasks. Correlatively, Figure 4 supports

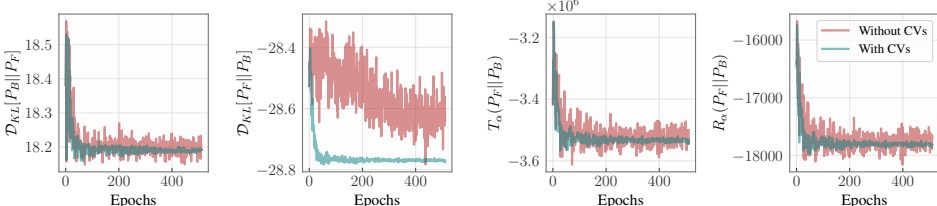

Figure 6: **Learning curves for different objective functions** in the task of set generation. The reduced variance of the gradient estimates notably increases training stability and speed.

the fact that minimizing divergence-based objectives frequently implies better coverage of the target's high-probability regions; the only exception is the (extremely sparse) hypergrid task [55].

### 5.3 Reducing the variance of the estimated gradients

Figure 2 demonstrates that implementing CVs for the REINFORCE estimator reduces the noise level of gradient estimates significantly. This reduction in variance also accelerates training convergence. To illustrate this, we use the same experimental setup from Section 5.2 and analyze the learning curves for each divergence measure with and without control variates.

**Results.** Figure 6 shows that the implemented gradient reduction techniques significantly enhance the learning stability of GFlowNets and drastically accelerate training convergence when minimizing the reverse KL divergence. Our results indicate that well-designed CVs for gradient estimation can greatly benefit GFlowNets training. Notably, similar improvements have been observed in the context of Langevin dynamics simulations [22, 31, 44] and policy gradient methods for RL [68, 100].

## 6 Conclusions, limitations and broader impact

**Discussion.** We showed in a comprehensive range of experiments that divergence measures common in VI — forward KL, reverse KL, Renyi-$\alpha$, and Tsallis-$\alpha$ — are effective learning objectives for training GFlowNets, performing competitively with or better than their balance-based counterparts. To achieve this, the introduction of efficacious control variates for low-variance gradient estimation of the divergence-based objectives was crucial, which is a key distinction between our work an prior art [55, 112]. Additionally, we developed the theoretical connection between GFlowNets and VI beyond the setting of finitely supported measures, establishing results for arbitrary topological spaces.

**Limitations.** Albeit comprehensive and on par with the wider literature, our empirical evaluation was performed on problems of relatively small size due to the intractability of probing a GFlowNet's distributional accuracy on very large state spaces. That said, we acknowledge that an assessment on the domains of natural language processing [30] and drug discovery [3] based on context-specific metrics would strengthen our conclusions; we leave these tasks to future endeavors. Similarly, while we observed promising results for $\alpha = 0.5$, there might be different choices of $\alpha$ that, depending on the application, might strike a better explore-exploit tradeoff and incur faster convergence. Thus, thoroughly exploring different $\alpha$ might be especially useful to practitioners.

**Broader impact.** Overall, our work highlights the potential of the once-dismissed VI-inspired schemes for training GFNs, paving the way for further research towards improving the GFlowNets by drawing inspiration from the VI literature. For instance, one could develop $\chi$-divergence-based losses for GFNs [19], combine GFNs with MCMC using Ruiz and Titsias [81]'s divergence, or employ an objective similar to that of importance-weighted autoencoders [10]. Finally, although an $\epsilon$-greedy off-policy sampling scheme can be easily incorporated into a divergence-minimizing algorithm through an importance-sampling correction, it remains elusive whether this would be possible for more sophisticated sampling techniques such as replay buffer [15] and local search [40].

## Acknowledgements

This work was supported by the Fundação Carlos Chagas Filho de Amparo à Pesquisa do Estado do Rio de Janeiro FAPERJ (SEI-260003/000709/2023), the São Paulo Research Foundation FAPESP (2023/00815-6), the Conselho Nacional de Desenvolvimento Científico e Tecnológico CNPq (404336/2023-0), and the Silicon Valley Community Foundation through the University Blockchain Research Initiative (Grant #2022-199610).

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

# A  Related works

**Generative Flow Networks.** GFlowNets [3, 4] were initially proposed as a reinforcement learning algorithm for finding diverse high-valued states in a discrete environment by sampling from a distribution induced by a reward function. Shortly after, they were extended to sample from complex distributions in arbitrary topological spaces [48]. Remarkably, this family of models was successfully applied to problems as diverse as structure learning and causal discovery [13, 15, 16], discrete probabilistic modeling and graphical models [24, 29, 107, 108], combinatorial optimization and stochastic control [109, 110], drug discovery [3, 33, 62], design of biological sequences [32], natural language processing [30], and aerial scene classification [23]. Concomitantly to these advances, there is a growing literature concerned with the development of more efficient training algorithms for GFlowNets [4, 40, 55, 84] — primarily drawing inspiration from existing techniques in the reinforcement learning literature [60, 65, 66, 87]. In the same spirit, Tiapkin et al. [87] showed it is possible to frame GFlowNets as an entropy-regularized reinforcement learning. In a study closely related to ours, Malkin et al. [56] proved the equivalence between GFlowNets and hierarchical variational inference (HVI) for discrete distributions; however, when training GFlowNets using divergence-based methods from the VI literature, the authors found no improvement relatively to the traditional flow-matching objectives. Thus, extending beyond discrete distributions, this work provides a definitive analysis of training GFlowNets by directly optimizing a set of divergences typically employed in variational inference training, given a clear context and conditions for effective use of divergence objectives for efficient learning procedures applied on GFlowNets models.

**Reinforcement Learning as Inference.** Reinforcement Learning (RL) has been studied as a form of probabilistic inference extensively, generating relevant insights in the literature, and alternatively referred to as *control as inference*. Todorov [88] demonstrates a duality between estimation and optimal control, establishing conditions where estimation algorithms could applied for control problems. Kappen et al. [38] demonstrated that optimal control problems could be framed as inference problems in graphical models, providing a unified perspective for solving control tasks. Levine [50] presents a complete and modern RL formulation, linking with VI in particular. Rudner et al. [80] integrates even further RL with VI methods, demonstrating the conceptual and algorithmic gains of leveraging outcome-driven RL with variational inference to optimize policy distributions. Developing further, Toussaint and Storkey [89] applies approximate probabilistic inference methods to solve Markov Decision Processes (MDPs) with discrete and continuous states. The approach also aligns with model-based RL techniques, such as *PILCO*, which utilizes probabilistic models to enhance data efficiency in policy search [14]. Recent work by Deleu et al. [17] positions discrete probabilistic inference as a control problem in multi-path environments, highlighting the synergy between control theory and probabilistic modeling in the context of GFlowNets. This body of works relates to the approach presented in this paper, comparing optimization of trajectory balance and flow-matching losses related to sequential decisions modeled by the GFlowNet with $f$-divergence measures minimization procedures – related to approximated variational inference and generalized posterior inference [36, 45, 51, 92, 95].

**Divergence measures and gradient reduction for VI.** Approximate inference via variational inference (VI) methods [6, 7, 36, 92] initially relied on message passing and coordinate ascent methods to minimize the KL divergence of an unnormalized distribution and a proposal in a parameterized tractable family of distributions. Despite the initial generality of the optimization perspective, the concrete implementation of algorithms often requires specialized update equations and learning objectives for specific classes of models. On the other hand, the development of algorithms and software for automatic differentiation [2] and stochastic gradient estimators [59] unlocked the potential application of generic gradient-based optimization algorithms in inference and learning tasks for a comprehensive class of models. Seminal works such as Black-Box VI (BBVI) [71], using the REINFORCE/score function estimator, and Automatic Differentiation VI (ADVI) [46], using reparameterization and change-of-variables, demonstrated practical algorithms for Bayesian inference in generic models, including models combining classical statistical modeling with neural networks. Overall, Mohamed et al. [59] explain the development of the main gradient estimators: the score function [11, 71, 97, 103], and the pathwise gradient estimator, also known as the parametrization trick [42, 43, 75]. The vanilla REINFORCE/score function estimator has notoriously high variance [11, 71, 77, 97], which prompted a body of work exploring variance reduction techniques. In the original BBVI proposal, Ranganath et al. [71] explored Rao-Blackwellization, combining iterated conditional expectations and control variates, using the score function estimator (given its zero

expectation) as a control variate. Subsequent works have continued to refine these techniques; Liu et al. [53] uses Rao-Blackwellized stochastic gradients for discrete distributions, while Kim et al. [39] and Wang et al. [93] explored joint control variates and provable linear convergence in BBVI. Additionally, Domke [21] and Domke [20] provided smoothness and gradient variance guarantees, further enhancing the robustness of score function estimator for VI methods. Our work demonstrates that effective variance reduction techniques applied to a $f$-divergence minimization training can significantly enhance the convergence speed and stability of the procedure. In theory and practice, we observed high compatibility between our results of variance-reduced $f$-divergence GFlowNets training and the body of work of variance-reduced score-function estimators for VI. Furthermore, by showing that these techniques apply to a broad class of models and optimization objectives, including continuous and mixed structured supports, we move GFlowNets' $f$-divergence minimization training closer to recent notions of generalized Bayesian inference and generalized VI[45] and variational inference in function spaces [95] – with the common thread of casting posterior inference as an optimization problem guided by some divergence measure. This generalization can enable applications of GFlowNets to a diverse range of machine learning tasks, enhancing their versatility and practical relevance.

# B Detailed description of the generative tasks

**Set generation** [3, 34, 65, 66]. $\mathcal{S}$ contains the sets of size up to $N$ with elements extracted from a fixed deposit $\mathcal{D}$ of size $D \geq N$ and $s_o = \emptyset$. For $s \in \mathcal{S}$ with $|s| < N$, $\kappa_f(s, \cdot)$ is a counting measure supported at (the $\sigma$-algebra induced by) $\{s \cup \{d\} \colon d \in \mathcal{D} \setminus s\}$; for $|s| = N$, $\kappa_f(s, \cdot) = \delta_{s_f}$. Then, $\mathcal{X} = \{s \in \mathcal{S} \colon |s| = N\}$. Similarly, $\kappa_b(s, \cdot)$'s support is $\{s \setminus \{d\} \colon d \in \mathcal{D}\}$ for $s \neq s_o$. We define the unnormalized target's density by $\log r(x) = \sum_{d \in x} f(d)$ for a fixed function $f \colon \mathcal{D} \to \mathbb{R}$. We parameterize $p_F(s, \cdot)$ as a deep set [105] and fix $p_B(s, \cdot)$ as a uniform density for $s \in \mathcal{S}$.

**Autoregressive sequence generation** [32, 55]. A *sequence* $s$ in $\mathcal{D}^n$, for any $K > n$, is bijectively associated to an element of $\mathcal{D} \times [K]$ by $s \mapsto \{(s_m, m) \colon 1 \leq m \leq n\} \cup \{(\perp, m) \colon K \geq m > n\}$; $\perp$ is a token denoting the sequence's end. In this context, we let $\mathcal{S} \subset \mathcal{P}(\mathcal{D} \times [N+1])$ be the set of sequences of size up to $N$, i.e., if $s \in \mathcal{S}$ and $(\perp, n+1) \in s$, then $(d, m) \in s$ iff $d = \perp$ for $n < m \leq N+1$ and there is $v \in (\mathcal{D} \cup \{\perp\})^n$ such that $(v_m, m) \in s$ for $m \leq n$; the initial state is $s_o = \emptyset$. For conciseness, we write $d \notin s$, meaning that $(d, i) \notin s$ for every $i$. Next, $\kappa_f(s, \cdot)$ is the counting measure supported at $\{s \cup \{(d, |s|+1)\} \colon d \in \mathcal{D} \cup \{\perp\}\}$ if $|s| < N$ and $\perp \notin s$; at $\{s \cup \{(\perp, N+1)\}\}$ if $|s| = N$; and at $\{s_f\}$ otherwise. Thus, $\mathcal{X} = \{s \in \mathcal{S} \colon \perp \in s\}$. Also, $k_b(s, \cdot)$ is supported at $\{s \setminus \{(d, |s|)\} \colon d \in \mathcal{D}\}$, which has a single element corresponding to the removal of the element of $s$ of the largest index. On the other hand, the target's density is $\log r(x) = \sum_{(d,i) \in x \colon d \neq \perp} f(d)g(i)$ for functions $f, g \colon \mathcal{D} \to \mathbb{R}$. We employ an MLP to parameterize $p_F(s, \cdot)$.

**Bayesian phylogenetic inference (BPI)** [111]. A (rooted) *phylogeny* is a complete binary tree with labeled leaves and weighted edges; each leaf corresponds to a biological species, and the edges' weights are a measurement of evolutionary time. Formally, we let $\mathcal{B}$ be the set of observed biological species and $\mathcal{V}_\mathcal{B}$ be the set of $|\mathcal{B}| + 1$ unobserved species. Next, we represent a phylogeny over $\mathcal{B}$ as a weighted directed tree $G_\mathcal{B} = (\mathcal{B} \cup \mathcal{V}_\mathcal{B}, E_\mathcal{B}, \omega_\mathcal{B})$ with edges $E_\mathcal{B}$ featured with a weight assignment $\omega_\mathcal{B}$; we denote its root by $r(G_\mathcal{B})$. Under these conditions, we define $\mathcal{S} = \left\{ \bigcup_{k=1}^{K} G_{\mathcal{F}_k} \colon \bigsqcup_k \mathcal{F}_k = \mathcal{B} \wedge \mathcal{G}_{\mathcal{F}_k} \text{ is a tree} \right\}$ as the set of forests built upon phylogenetic trees over disjoint subsets of $\mathcal{B}$; $\bigsqcup$ represents a disjoint union of non-empty sets and $s_o = \bigcup_{b \in \mathcal{B}} G_{\{b\}}$ is the forest composed of singleton trees $G_{\{b\}}$. Also, we define the *amalgamation* of phylogenies $G_{\mathcal{F}_k}$ and $G_{\mathcal{F}_j}$, $\mathcal{A}(G_{\mathcal{F}_k}, G_{\mathcal{F}_j})$, as the tree obtained by joining their roots $r(G_{\mathcal{F}_k})$ and $r(G_{\mathcal{F}_j})$ to a new node $r(G_{\mathcal{F}_k} \cup G_{\mathcal{F}_j})$, with a corresponding extension of the edges' weights. In contrast, the *dissolution* of a tree $G_\mathcal{F}$, $\mathcal{R}(G_\mathcal{F})$, returns the union of the two subtrees obtained by removing $r(G_\mathcal{F})$ from $G_\mathcal{F}$. Then, $\kappa_f(s, \cdot)$ is the counting measure supported at $\left\{ \bigcup_{k=1, k \neq i,j}^{K} G_{\mathcal{F}_k} \cup \mathcal{A}(G_{\mathcal{F}_i}, G_{\mathcal{F}_j}) \colon (i, j) \in [K]^2, i \neq j \right\}$ with $s = \bigcup_{k=1}^{K} G_{\mathcal{F}_k}$ and $K \geq 2$; if $s = \mathcal{G}_\mathcal{B}$, $\kappa_f(s, \cdot) = \delta_{s_f}$. Hence, $\mathcal{X}$ is the set of phylogenies over $\mathcal{B}$. Likewise, $\kappa_b(s, \cdot)$'s support is $\left\{ \bigcup_{k=1, k \neq j}^{K} G_{\mathcal{F}_k} \cup \mathcal{R}(G_{\mathcal{F}_i}) \colon i \in [K] \wedge r(G_{\mathcal{F}_i}) \notin \mathcal{B} \right\}$ for $s = \bigcup_{k=1}^{K} G_{\mathcal{F}_k}$ and $K \leq |\mathcal{B}|$. Finally, the unnormalized target is the posterior distribution defined by JC69's mutation model [37] given a data set of genome sequences of the species in $\mathcal{B}$. More specifically, we let the prior be a uniform distribution and compute the model-induced likelihood function by the efficient Felsenstein's algorithm [25]. We assume the edges' weights are constant. See [101] for further details. We parameterize $p_F(s, \cdot)$ with GIN [99] and fix $p_B(s, \cdot)$ as an uniform distribution.

**Mixture of Gaussians** [48, 110]. The training of GFlowNets in continuous spaces is challenging, and the problem of designing highly expressive models in this setting is still unaddressed [16, 48]. However, as we show in Section 5.2, divergence-based measures seem to be very effective learning objectives for autoregressive sampling of a sparse mixture of Gaussians. For a $d$-dimensional Gaussian distribution, $\mathcal{S} = \{\{(0,0), (x_i, i) \colon 1 \leq i \leq n\}, \colon n \leq d, x \in \mathbb{R}^n\} \subset \{(0,0)\} \cup \mathcal{P}(\mathbb{R} \times [d])$ and $s_o = (0, 0)$; note $\mathcal{S}$ is isomorphic to $\mathbb{R}^d$. Also, for $s = \{(x_i, i)\}_{i=1}^{n}$, $\kappa_f(s, \cdot)$ is Lebesgue's measure at $\{s \cup (x, n+1) \colon x \in \mathbb{R}\}$ if $n < d$ and $\kappa_f(s, \cdot) = \delta_{s_f}$ otherwise. In particular, $\mathcal{X} = \{s \in \mathcal{S} \colon \max_{(x,i) \in s} i = d\}$. Moreover, $\kappa_b(s, \cdot)$ is a measure on $\{s \setminus (x, |s|) \colon x \in \mathbb{R}\}$, which is isomorphic to $\mathbb{R}$, which is a singleton. We define the target's density with a homogeneous mixture of Gaussian distributions, $\frac{1}{K} \sum_{i=1}^{K} \mathcal{N}(\mu_i, \sigma^2 I)$ with $\mu_i \in \mathbb{R}^d$. We similarly define $P_F(s, \cdot)$ as a mixture of one-dimensional Gaussians with mean and variance learned via an MLP [48].

**Banana distribution.** [57, 76] We consider sampling from the banana distribution, defined by

$$\mathcal{N} \left( \begin{bmatrix} x_1 \\ x_2 + x_1^2 + 1 \end{bmatrix} \middle| \begin{bmatrix} 0 \\ 0 \end{bmatrix}, \begin{bmatrix} 1 & 0.9 \\ 0.9 & 1 \end{bmatrix} \right). \tag{8}$$

Given its geometry and shape, this distribution is a common baseline in the approximate Bayesian inference literature [90, 104, 106]. This task is identical to sampling from a mixture of Gaussian distributions, except for the different target density specified by the model in Equation (8). Also, we rely on the implementation Hamiltonian Monte Carlo (HMC) [5, 61] provided by Stan [12] to obtain accurate samples from (8).

A similar description may be utilized for the hypergrid and structure learning domains.

# C  Proofs

We will consider the measurable space of *trajectories* $(\mathcal{P}_\mathcal{S}, \Sigma_P)$, with $\mathcal{P}_\mathcal{S} = \{(s, s_1, \ldots, s_n, s_f) \in \mathcal{S}^{n+1} \times \{s_f\} \colon 0 \le n \le N - 1\}$ and $\Sigma_P$ as the $\sigma$-algebra generated by $\bigcup_{n=1}^{N+1} \Sigma^{\otimes n}$. For notational convenience, we use the same letters for representing the measures and kernels of $(\mathcal{S}, \Sigma)$ and their natural product counterparts in $(\mathcal{P}_\mathcal{S}, \Sigma_P)$, which exist by Carathéodory extension's theorem [96]; for example, $\nu(B) = \nu^{\otimes n}(B)$ for $B = (B_1, \ldots, B_n) \in \Sigma^{\otimes n}$ and $p_{F_\theta}(\tau|s_o; \theta)$ is the density of $P_F^{\otimes n+1}(s_o, \cdot)$ for $\tau = (s_o, s_1, \ldots, s_n, s_f)$ relatively to $\mu^{\otimes n}$. In this case, we will write $\tau$ for a generic element of $\mathcal{P}_\mathcal{S}$ and $x$ for its terminal state (which is unique by Definition 1).

## C.1  Proof of Proposition 1

We will show that the gradient of the expected on-policy TB loss matches the gradient of the KL divergence between the forward and backward policies. Firstly, note that

$$
\begin{aligned}
\nabla_\theta \mathcal{D}_{KL}[P_F || P_B] &= \nabla_\theta \mathbb{E}_{\tau \sim P_F(s_o, \cdot)} \left[ \log \frac{p_F(\tau|s_o; \theta)}{p_B(\tau)} \right] \\
&= \nabla_\theta \int_\tau \log \frac{p_F(\tau|s_o; \theta)}{p_B(\tau)} \mathrm{d}P_F(s_o, \mathrm{d}\tau) \\
&= \nabla_\theta \int_\tau \log \frac{p_F(\tau|s_o; \theta)}{p_B(\tau)} p_F(\tau|s_o; \theta) \mathrm{d}\kappa_f(s_o, \mathrm{d}\tau) \\
&= \int_\tau \nabla_\theta \log \frac{p_F(\tau|s_o; \theta)}{p_B(\tau)} P_F(s_o, \mathrm{d}\tau) \\
&\quad + \int_\tau \log \frac{p_F(\tau|s_o; \theta)}{p_B(\tau)} \nabla_\theta p_F(\tau|s_o; \theta) \mathrm{d}\kappa_f(s_o, \mathrm{d}\tau)
\end{aligned}
$$

by Leibniz's rule for integrals and the product rule for derivatives. Then, since $\nabla_\theta f(\theta) = f(\theta) \nabla \log f(\theta)$ for any differentiable function $f \colon \theta \mapsto f(\theta)$,

$$
\nabla_\theta \mathcal{D}_{KL}[P_F || P_B]
$$

$$
= \mathbb{E}_{\tau \sim P_F(s_o, \cdot)} \left[ \nabla_\theta \log p_F(\tau|s_o) + \log \frac{p_F(\tau|s_o)}{p_B(\tau)} \nabla_\theta \log p_F(\tau|s_o) \right] \tag{9}
$$

$$
= \mathbb{E}_{\tau \sim P_F(s_o, \cdot)} \left[ \log \frac{p_F(\tau|s_o)}{p_B(\tau)} \nabla_\theta \log p_F(\tau|s_o) \right];
$$

we omitted the dependency of $P_F$ (and of $p_F$ thereof) on the parameters $\theta$ for conciseness. On the other hand,

$$
\nabla_\theta \mathcal{L}_{TB}(\tau; \theta) = 2 \left( \log \frac{p_F(\tau|s_o; \theta)}{p_B(\tau)} \right) \nabla_\theta \log p_F(\tau) \tag{10}
$$

by the chain rule for derivatives. Thus,

$$
\mathbb{E}_{\tau \sim P_F(s_o, \cdot)} \nabla_\theta \mathcal{L}_{TB}(\tau; \theta) = 2 \nabla_\theta \mathcal{D}_{KL}[P_F || P_B], \tag{11}
$$

ensuring that the equivalence between $\mathcal{L}_{TB}$ and $\mathcal{D}_{KL}$ in terms of expected gradients holds in a context broader than that of finitely supported distributions [56].

## C.2  Proof of Lemma 1

Henceforth, we will recurrently refer to the score estimator for gradients of expectations [97], namely,

$$
\nabla_\theta \mathbb{E}_{\tau \sim P_F(s_o, \cdot)} [f_\theta(\tau)] = \mathbb{E}_{\tau \sim P_F(s_o, \cdot)} [\nabla_\theta f_\theta(\tau) + f_\theta(\tau) \nabla_\theta \log p_F(\tau|s_o; \theta)], \tag{12}
$$

which can be derived using the arguments of the preceding section. In this context, the Renyi-$\alpha$'s divergence satisfies

$$
\nabla_\theta R_\alpha(P_F || P_B) = \frac{\nabla_\theta \mathbb{E}_{\tau \sim P_F(s_o, \cdot)} [g(\tau, \theta)]}{(\alpha - 1) \mathbb{E}_{\tau \sim P_F(s_o, \cdot)} g(\tau, \theta)},
$$

with $g(\tau; \theta) = \left( p_B(\tau|x) r(x) / p_F(\tau|s_o; \theta) \right)^{1-\alpha}$ and $\alpha \ne 1$; similarly, the Tsallis-$\alpha$'s divergence abides by

$$
\nabla_\theta T_\alpha(P_F || P_B) = \frac{1}{(\alpha - 1)} \nabla_\theta \mathbb{E}_{\tau \sim P_F(s_o, \cdot)} [g(\tau, \theta)]. \tag{13}
$$

The statement then follows by substituting $\nabla_\theta \mathbb{E}_{\tau \sim P_F(s_o, \cdot)}[g(\tau, \theta)]$ with the corresponding score estimator given by Equation (12).

## C.3 Proof of Lemma 2

**Forward KL divergence.** The gradient of $\mathcal{D}_{KL}[P_B||P_F]$ is straightforwardly obtained through the application of Leibniz's rule for integrals,

$$\nabla_\theta \mathcal{D}_{KL}[P_B||P_F] = -\mathbb{E}_{\tau \sim P_B(s_f, \cdot)}[\nabla_\theta \log p_F(\tau|s_o; \theta)],$$

since the averaging distribution $P_B$ do not depend on the varying parameters $\theta$. However, as we compute Monte Carlo averages over samples of $P_F$, we apply an importance reweighting scheme [63, Chapter 9] to the previous expectation to infer that, up to a positive multiplicative constant,

$$\nabla_\theta \mathcal{D}_{KL}[P_B||P_F] \stackrel{C}{=} -\mathbb{E}_{\tau \sim P_F(s_o, \cdot)}\left[\frac{p_B(\tau|x)r(x)}{p_F(\tau|s_o; \theta)}\nabla_\theta \log p_F(\tau|s_o; \theta)\right],$$

with $\stackrel{C}{=}$ denoting equality up to a positive multiplicative constant. We emphasize that most modern stochastic gradient methods for optimization, such as Adam [41] and RMSProp [28], remain unchanged when we multiply the estimated gradients by a fixed quantity; thus, we may harmlessly compute gradients up to multiplicative constants.

**Reverse KL divergence.** We verified in Equation (9) that

$$\nabla_\theta \mathcal{D}_{KL}[P_F||P_B] = \mathbb{E}_{\tau \sim P_F(s_o, \cdot)}\left[\log \frac{p_F(\tau|s_o)}{p_B(\tau)}\nabla_\theta s_\theta(\tau)\right].$$

Since $p_B(\tau) = p_B(\tau|x)r(x)/Z$ and $\mathbb{E}_{\tau \sim P_F(s_o, \cdot)}\nabla_\theta s_\theta(\tau) = 0$, the quantity $\nabla_\theta \mathcal{D}_{KL}[P_F||P_B]$ may be rewritten as

$$\begin{aligned}
\nabla_\theta \mathcal{D}_{KL}[P_F||P_B] &= \mathbb{E}_{\tau \sim P_F(s_o, \cdot)}\left[\log \frac{p_F(\tau|s_o)}{p_B(\tau)}\nabla_\theta s_\theta(\tau)\right] \\
&\quad + \mathbb{E}_{\tau \sim P_F(s_o, \cdot)}[(\log Z)\nabla_\theta s_\theta(\tau)] \\
&= \mathbb{E}_{\tau \sim P_F(s_o, \cdot)}\left[\log \frac{p_F(\tau|s_o)}{p_B(\tau|x)r(x)}\nabla_\theta s_\theta(\tau)\right],
\end{aligned}$$

in which $x$ is the terminal state corresponding to the trajectory $\tau$. Thus proving the statement in Lemma 2.

## C.4 Proof of Proposition 2

We will derive an expression for the optimal baseline of a vector-valued control variate. For this, let $f$ be the averaged function and $g: \tau \mapsto g(\tau)$ be the control variate. Assume, without loss of generality, that $\mathbb{E}_\pi[g] = 0$ for the averaging distribution $\pi$ over the space of trajectories. In this case, the optimal baseline for the control variate $a^\star$ is found by

$$a^\star = \arg\min_{a \in \mathbb{R}} \text{Tr}\left(\text{Cov}_{\tau \sim \pi}[f(\tau) - a \cdot g(\tau)]\right). \tag{14}$$

Thus,

$$a^\star = \arg\min_{a \in \mathbb{R}} \text{Tr}\left(-2a \cdot \text{Cov}_\pi[f(\tau), g(\tau)] + a^2\text{Cov}_\pi(g(\tau))\right),$$

which is a convex optimization problem solved by

$$\begin{aligned}
a^\star &= \frac{\text{Tr}\left(\text{Cov}_\pi[f(\tau), g(\tau)]\right)}{\text{Tr}\left(\text{Cov}_\pi[g(\tau)]\right)} \\
&= \frac{\text{Tr}\,\mathbb{E}_\pi[(f - \mathbb{E}_\pi[f(\tau)])g(\tau)^T]}{\text{Tr}\,\mathbb{E}_\pi[g(\tau)g(\tau)^T]} \\
&= \frac{\mathbb{E}_\pi[\text{Tr}\,(f - \mathbb{E}_\pi[f(\tau)])^T g(\tau)]}{\mathbb{E}_\pi[\text{Tr}\,g(\tau)^T g(\tau)]} \\
&= \frac{\mathbb{E}_\pi[(f - \mathbb{E}_\pi[f(\tau)])^T g(\tau)]}{\mathbb{E}_\pi[g(\tau)^T g(\tau)]},
\end{aligned} \tag{15}$$

in which we used the circular property of the trace. This equation exactly matches the result in Proposition 2. In practice, we use $g(\tau) = \nabla_\theta \log p_F(\tau)$ for both the reverse KL- and $\alpha$-divergences, rendering a baseline $a^\star$ that depends non-linearly on the sample gradients and is hence difficult to compute in a GPU-powered autodiff framework efficiently. We thus use Equation (6) to estimate $a^\star$.

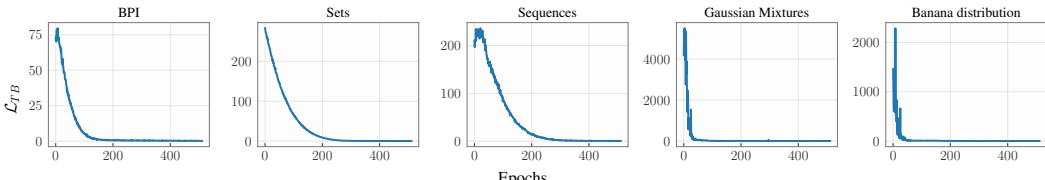

Figure 7: **Learning curves for a GFlowNet trained by minimizing the TB loss.** The curves' smoothness highlights the low variance of the optimization steps incurred by the stochastic gradients of $\mathcal{L}_{TB}$, which do *not* use a score function estimator.



Figure 9: **Additional illustration of the effect of** $\alpha$ when learning GFlowNets by minimizing the Renyi-$\alpha$ divergence in the hypergrid environment. For such a sparse target distribution, a large and negative value of $\alpha$ (left) is beneficial to ensure that all modes are appropriately covered by the learned distribution. In contrast, the mode-seeking behavior induced by a large value of $\alpha$ entails the collapse of the model in a single mode (right).

## D Additional experiments

**Gradient variance for flow-network-based objectives.** Figure 7 shows the learning curve for the TB loss in each of the generative tasks. Notoriously, it underlines the low variance of the optimization steps — which, contrarily to their divergence-based counterparts, do not rely on a score function estimator — and suggests that the design of control variates for estimating the gradients of these objectives would not be significantly helpful. Also, the gradient of $\mathcal{L}_{TB}$ depends non-linearly on the score function $\log p_F$ and, consequently, it is unclear how to implement computationally efficient variance reduction techniques in this case.

**Forward KL for sequence generation.** Figure 3 shows that alternative approaches in terms of convergence speed outperformed a GFlowNet trained to minimize the forward KL. One possible cause of this underperformance is the high variance induced by the underlying importance sampling estimator. To verify this, we re-run the corresponding experiments, increasing the size of the batch of trajectories for the forward KL estimator to 1024. Figure 8 presents the experiment's results, with an increased batch size corresponding to an estimator of smaller variance that accelerates the GFlowNet's training convergence. More broadly, this suggests that the design of GFlowNet-specific variance reduction techniques, which we leave to future endeavors, may further improve this family of models.

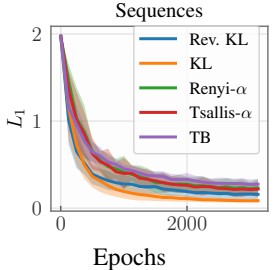

Figure 8: Results for sequence generation with larger batches.

**Influence of** $\alpha$ **on the learning of GFlowNets.** As mentioned earlier, divergence-based measures fall short compared to their balance-based counterparts for the hypergrid navigation task. For this extremely sparse problem, the benefits from off-policy exploration during training seem to supersede the statistical efficiency enacted by the minimization of divergences, which fail to properly cover the high-probability regions of the target distribution. In this scenario, Figure 9 suggests that this issue can be mildly circumvented by choosing a sufficiently negative $\alpha$ for the Renyi-$\alpha$ divergences, thereby imposing a mass-covering behavior to the learned model. However, these results should be substantiated by further experimental analysis to be conclusive. Currently, we would suggest a practitioner to stick to the balance-based objectives when dealing with very sparse target distributions.

# E    Experimental details

The following paragraph provides further implementation details. Regarding open access to the code, we will make the code public upon acceptance.

**Shared configurations.** For every generative task, we used the Adam optimizer [41] to carry out the stochastic optimization, employing a learning rate of $10^{-1}$ for $\log Z_\theta$ when minimizing $\mathcal{L}_{TB}$ and $10^{-3}$ for the remaining parameters, following previous works [48, 55, 56, 66]. We polynomially annealed the learning rate towards 0 along training, similarly to [86]. Also, we use LeakyReLU [98] as the non-linear activation function of all implemented neural networks.

**Set generation.** We implement an MLP of 2 64-dimensional layers to parameterize the policy's logits $\log p_F(s, \cdot)$. We train the model for 512 epochs with a batch of 128 trajectories for estimating the gradients. Also, we let $D = 32$ and $N = 16$ be the source's and set's sizes, respectively.

**Autoregressive sequence generation.** We parameterize the logits of the forward policy with a MLP of 2 64-dimensional layers; we pad the sequences to account for their variable sizes. We respectively consider $D = 8$ and $N = 6$ for the source's and sequence's sizes. To approximate the gradients, we rely on a batch of 128 sequences.

**Bayesian phylogenetic inference.** We parameterize the logits of the forward policy with a 2-layer GIN [99] with a 64-dimensional latent embedding, which is linearly projected to $\log p_F$. Moreover, we simulated the JC69 model [37] to obtain 25-sized sequences of nucleotides for each of the 7 observed species, setting $\lambda = 0.3$ for the instantaneous mutation rate; see [101] for an introduction to computational phylogenetics and molecular evolution. To estimate the gradients, we relied on batches of 64 trajectories.

**Hypergrid navigation.** We consider a $H = 12$ for Figure 3 and Figure 4 and $H = 9$ for Figure 9. In both cases, $d = 2$ and $R_o = 10^{-3}$; see [55, Section 5.1]. To parameterize the policy, we used a 2-layer 256-dimensional MLP with ReLU activations. We trained the models for 10000 epochs.

**Bayesian structure learning.** We simulated a 100-sized 5-variable data set $\mathbf{X}$ from a randomly parameterized linear Gaussian structural model based on a fixed Bayesian network. We implemented a 2-layer 256-dimensional MLP with ReLU activations for the policy network, which received the flattened adjacency matrix of the current state as input. Training lasted for 4000 epochs.

**Mixture of Gaussian distributions.** We consider a mixture of 9 2-dimensional Gaussian distributions centered at $\mu_{ij} = (i, j)$ for $0 \le i, j \le 2$, each of which having an isotropic variance of $10^{-1}$; see Figure 1. We use an MLP of 2 64-dimensional layers to parameterize the forward policy.

**Banana-shaped distribution.** The model is specified by Equation (8). We also consider an MLP of 2 64-dimensional layers to parameterize the forward policy.

