# OpenReview forum: "On Divergence Measures for Training GFlowNets"
_NeurIPS.cc/2024/Conference — NeurIPS 2024 poster_

### Official Review · Reviewer_Gn1e · 2024-06-17

**Soundness:** 2
**Presentation:** 2
**Contribution:** 2
**Rating:** 5
**Confidence:** 4

**Summary:**

This paper investigates alternative training methods for Generative Flow Networks (GFlowNets) by evaluating various divergence measures, including Renyi-α, Tsallis-α, reverse, and forward Kullback-Leibler (KL) divergences. Traditional methods focusing on minimizing log-squared differences are shown to lead to biased and high-variance estimators. The authors propose efficient estimators for the stochastic gradients of these divergences and introduce control variates to reduce gradient variance. Empirical results across diverse tasks demonstrate that minimizing these divergence measures significantly accelerates training convergence and enhances stability compared to traditional methods. The paper establishes theoretical connections between GFlowNets and variational inference, extending these insights to arbitrary topological spaces, and highlights the practical effectiveness of control variates in improving training efficiency.

**Strengths:**

- **Comprehensive Evaluation**: The paper thoroughly evaluates multiple divergence measures and their impact on GFlowNet training.
- **Theoretical Insights**: Establishing theoretical connections between GFlowNets and VI broadens the understanding of these models.
- **Practical Contributions**: The design of control variates to reduce gradient variance is a significant practical contribution that can be applied in various optimization scenarios.
- **Variance Reduction Techniques**: The introduction of variance reduction techniques, including control variates and leave-one-out estimators, effectively addresses the high variance issue in gradient estimates, enhancing the learning stability and efficiency.

**Weaknesses:**

- **Theory** The theoretical contribution builds upon previous works [4,5]. However, I do not think the theory itself has enough contribution. Because the Measurable pointed DAG is from previous work [5] and the main theoretical claim, Proposition 1, has limited novelty compared with Proposition 1 in [4]. Therefore, unlike previous works [4,5], they have their novel theoretical contributions and synthetic experiments would suffice. Thus, for this paper, stronger experiments are required, as will be discussed in the next.
- **Experiments - Part 1** This paper does not contain enough real-world tasks, such as fragment-based molecule generation [1], graph combinatorial optimization [2], and RNA sequence generation [3] to illustrate its main contribution. These are standard tasks in evaluating GFN performances and are necessary to include. For the only real-world task in the paper, i.e., the BPI task, this paper admits that "not statistically significant" in line 334. Therefore, it is unclear whether the proposed methods to use other divergence measures will be meaningful in real-world scenarios.
- **Experiments - Part 2** For the synthetic tasks, the plots in Figure 3 are also missing some lines. For example, why are there only two lines in the **Sets** plots? Also, there is no single best loss function that is uniformly better than the KL baseline. Therefore, additional divergence might seem unnecessary. There should be stronger evidence to support the use of other measures. Or, the authors can provide a guideline on how to choose the best measures given prior information about the tasks.

- I would raise my scores if additional experiments are included and the proposed methods are indeed beneficial in more complex tasks.


[1] Jain, Moksh, et al. "Biological sequence design with gflownets." International Conference on Machine Learning. PMLR, 2022.

[2] Zhang, Dinghuai, et al. "Let the flows tell: Solving graph combinatorial problems with GFlowNets." Advances in Neural Information Processing Systems 36 (2024).

[3] Kim, Minsu, et al. "Local search gflownets." arXiv preprint arXiv:2310.02710 (2023).

[4] Malkin, Nikolay, et al. "GFlowNets and variational inference." arXiv preprint arXiv:2210.00580 (2022).

[5] Lahlou, Salem, et al. "A theory of continuous generative flow networks." International Conference on Machine Learning. PMLR, 2023.

**Questions:**

See **Weaknesses**.

**Limitations:**

See **Weaknesses**.

---

> ### Author Rebuttal · Authors · 2024-08-07
>
> We thank the reviewer for their detailed feedback and suggestions. Below, we address the specific weaknesses and questions.  We hope our clarifications and additional experiments address your concerns and elevate your appraisal of our work.
> .
> > Because the Measurable pointed DAG is from previous work [5] and the main theoretical claim, Proposition 1, has limited novelty compared with Proposition 1 in [4].
>
> We would like to emphasize Proposition 1 is not the main contribution of our work. In fact, note this proposition is part of our background section. While perhaps a simple extension of previous results, Proposition 1 is a necessary formalism to our developments in the following sections, as we consider discrete as well as continuous state spaces in our work.
>
> > This paper does not contain enough real-world tasks, such as fragment-based molecule generation [1], graph combinatorial optimization [2], and RNA sequence generation [3] to illustrate its main contribution. These are standard tasks in evaluating GFN performances and are necessary to include. For the only real-world task in the paper, i.e., the BPI task, this paper admits that "not statistically significant" in line 334. Therefore, it is unclear whether the proposed methods to use other divergence measures will be meaningful in real-world scenarios.
>
> We would like to emphasize that, before our work, divergence measures were perceived as inferior choices to traditional GFlowNet criteria (e.g., TB) as suggested by Malkin et al. [4]. The main reason for our success is the design of appropriate control variates (CVs) for variance reduction, which preserve the unbiasedness of gradient estimates. It is important to note that Malkin et al. [4]'s gradient estimators differ from ours — and are not bias-free, as we discuss in lines 272-278 of our manuscript.
>
> To strengthen our empirical claims, we have included additional baselines (DB, VarInf and SubTB) [8, 9, 10, 11] and environments (hypergrid and causal structure learning) [4, 6, 7] — see Figure 2 in the rebuttal PDF. It is worth mentioning that causal structure learning is one of the prime applications of GFlowNets [6, 7] and can be seen as an instance of combinatorial optimization. Our results reinforce that (given appropriate control variables) divergence-based objectives perform similarly to or better than balance-based objectives.
>
> Please let us know if this addresses your issue. If you believe considering additional environments is necessary to verify our claim, we would gladly run more experiments in the discussion period.
>
> > For the synthetic tasks, the plots in Figure 3 are also missing some lines. For example, why are there only two lines in the Sets plots? Also, there is no single best loss function that is uniformly better than the KL baseline. Therefore, additional divergence might seem unnecessary. There should be stronger evidence to support the use of other measures. Or, the authors can provide a guideline on how to choose the best measures given prior information about the tasks.
>
> We apologize for the oversight in Figure 3. The missing lines in the plot for the set generation task were due to the overlap of learning curves for the divergence-based objectives. We will include dashed lines to improve the visualization.
>
> We highlight that we are considering four divergence measures: Forward-KL, Reverse-KL, $\alpha$-Renyi, and $\alpha$-Tsallis, obtaining for each a variance-reduced gradient estimators for GFlowNet training. These estimators are distinct from the proposal in Malkin et al. [4].
>
> We considered the TB loss as a baseline. We have also run comparisons against the DB, SubTB, and VarGrad losses for the rebuttal (Figs 1 and 2, rebuttal PDF). Compared to the baseline losses, we can always find a divergence loss with better performance. On the other hand, no single divergence has uniform dominance over the others (Reverse-KL and Forward-KL are distinct measures).
> Overall, the experimental evidence favors the class of divergence measures. One practical advantage of Renyi and Tsallis is to represent intermediate measures between the extremes of mode-seeking and mass-covering of Forward-KL and Reverse-KL. We have added experiments exploring the effect of $\alpha$ for the distribution of Hypergrid (Fig 3, rebuttal PDF).
>
> Regarding how to choose the most appropriate divergence, we agree that clear guidelines would be useful. However, developing these guidelines an open problem in the VI literature [12, 13].
>
> [1] Biological sequence design with gflownets. ICML 2022
>
> [2] Let the flows tell: Solving graph combinatorial problems with GFlowNets. NeurIPS 2024
>
> [3] Local search gflownets. arXiv 2023
>
> [4] GFlowNets and variational inference.  ICLR 2023
>
> [5] A theory of continuous generative flow networks. ICML 2023
>
> [6] Bayesian structure learning with generative flow networks. In UAI, 2022
>
> [7] Joint Bayesian Inference of Graphical Structure and Parameters with a GFlowNet. NeurIPS 2023
>
> [8] Learning GFlowNets from partial episodes for improved convergence and stability. ICML 2023
>
> [9] GFlowNet Foundations. JMLR 2023
>
> [10] VarGrad: A Low-Variance Gradient Estimator for VI. NeurIPS 2020
>
> [11] Robust Scheduling with GFlowNets. ICLR 2023
>
> [12]  Divergence measures and message passing. 2005
>
> [13] Rényi Divergence Variational Inference. NeurIPS 2016

---

> > ### Comment · Reviewer_Gn1e · 2024-08-09
> >
> > Thanks for your clarifications. I think the design of variance reduction in divergence-based objectives is a quite novel contribution. Admittedly, the design of appropriate divergence is an open question in VI literature. However, since the current experiments still do not include real-world environment like molecules, I can only raise my score to 5.

---

> > > ### Author Response · Authors · 2024-08-10
> > >
> > > Thank you very much for engaging in the discussion and acknowledging our rebuttal and the novelty of our work.

---

### Official Review · Reviewer_3sBX · 2024-07-12

**Soundness:** 3
**Presentation:** 4
**Contribution:** 3
**Rating:** 7
**Confidence:** 4

**Summary:**

GFlowNets are a probabilistic framework for training amortized samplers for high-dimensional compositional spaces. The samplers are typically trained using local consistency objectives which are squared log losses. This paper examines alternatives to these obejctives in the form of general statistical f-divergence measures. The authors consider forward and reverse KL, Renyi-$\alpha$ and Tsallis-$\alpha$ divergences. The authors derive gradients for the divergences in the case of a fixed $p_B$ and on-policy samples. These gradient computations rely on REINFORCE-style estimators and consequently can suffer from high-variance gradient estimates which affect the optimization procedure. For variance reduction, the authors derive control variates for their gradient estimators. Through a series of experiments on a variety of tasks, the authors demonstrate the effectiveness of these divergences for training GFlowNets.

**Strengths:**

* The paper is quite well written and clear. The authors are thorough and clear in introducing the central ideas and provide sufficient details making it easy to follow.
* The paper studies the important problem of finding the "right" learning objective in the context of training GFlowNets. Following a long line of work in VI, the authors leverage statistical divergences and propose optimizing them directly to learn the GFN sampler.
* Additionally, the authors improve upon prior work drawing a connection between GFNs and VI by proposing principled control variates to reduce the variance in the gradient estimates for the divergences from the REINFORCE estimators. The CVs seem to have a significant effect on the performance.
* The empirical analysis covers a variety of problems including continuous and discrete spaces as wells as general DAGs and tree spaces in the case of discrete spaces.
* The authors also include code with the submission aiding reproducibility.

**Weaknesses:**

* The paper considers the on-policy setting for training GFlowNets and the proposed learning objectives based on the divergences assume on-policy smaples. However, existing flow-based objectives are all off-policy, and the advantage of GFlowNets on a lot of tasks (specifically challenging tasks with multi-modal target distributions) comes from the ability to train on off-policy trajectories (e.g. replay buffer[1] to local search [2]). So while the proposed learning objectives empirically perform better than TB on tasks where on-policy sampling is enough, it lacks the flexibility of accomodating off-policy training.
* The authors also assume that the backward policy $P_B$ is fixed (L153). While this is true in some cases, learning $P_B$ results in significant improvements to the learned sampler[3,4]. As far as I can tell, modifying the proposed objectives to accomodate learning the P_B is non-trivial.
* I appreciate the diversity of problems studied by the authors in their experiments, but there are some gaps in the empirical analysis. Specifically, the authors only include a TB baseline and not other objectives such as SubTB, DB which can perform better than TB (and have better training stability) in some cases. Additionally, the paper also does not include the VarGrad-style objective [5] which does away the need for estimating $Z$ in TB.
* Moreover, the tasks consider relatively small tasks so it is not clear how scalable the proposed objectives are.


[1] Towards Understanding and Improving GFlowNet Training. Max W. Shen, Emmanuel Bengio, Ehsan Hajiramezanali, Andreas Loukas, Kyunghyun Cho, Tommaso Biancalani. ICML 2023.

[2] Local Search GFlowNets. Minsu Kim, Taeyoung Yun, Emmanuel Bengio, Dinghuai Zhang, Yoshua Bengio, Sungsoo Ahn, Jinkyoo Park. ICLR 2024.

[3] Trajectory balance: Improved credit assignment in GFlowNets. Nikolay Malkin, Moksh Jain, Emmanuel Bengio, Chen Sun, Yoshua Bengio. NeurIPS 2022.

[4] A theory of continuous generative flow networks. Salem Lahlou, Tristan Deleu, Pablo Lemos, Dinghuai Zhang, Alexandra Volokhova, Alex Hernández-García, Léna Néhale Ezzine, Yoshua Bengio, Nikolay Malkin. ICML 2023.

[5] Robust Scheduling with GFlowNets. David W. Zhang, Corrado Rainone, Markus Peschl, Roberto Bondesan. ICLR 2023.

**Questions:**

In addition to the points in Weaknesses:

* L135 says TB requires estimating $Z$ but KL doesn't but I am not sure that is correct? Since even in the KL you need the normalizing constant in the $P_B$ term.

* L146: [1] would be a more appropriate reference here I think?

* L157: Missing reference to [2]

* What is the computational performance (in terms of runtime) of the proposed objectives relative to TB?


[1] Rubinstein, R. Y. (1981). Simulation and the Monte Carlo Method. In Wiley Series in Probability and Statistics. Wiley. https://doi.org/10.1002/9780470316511

[2] f-Divergence Variational Inference. Neng Wan, Dapeng Li, Naira Hovakimyan. https://arxiv.org/abs/2009.13093.

**Limitations:**

The authors do not explicitly address the limitations of their approach (discussed in the weaknesses) section, though the assumptions are mentioned briefly in Section 2 and 3. The authors only mention the choice of $\alpha$ as a limitation.

There is also no discussion of broader impacts (the reference in the checklist is broken too).

---

> ### Author Rebuttal · Authors · 2024-08-06
>
> Thank you for the suggestions and for appreciating our work. We did our best to address each of your concerns below. Please let us know if you have other questions or require further clarification.
>
> > The paper considers the on-policy setting for training GFlowNets and the proposed learning objectives based on the divergences assume on-policy smaples. However, existing flow-based objectives are all off-policy, and the advantage of GFlowNets on a lot of tasks [...] comes from the ability to train on off-policy trajectories (e.g. replay buffer [1] to local search [2]). So while the proposed learning objectives empirically perform better than TB on tasks where on-policy sampling is enough, it lacks the flexibility of accomodating off-policy training.
>
> Indeed, many heuristics for off-policy learning of GFlowNets cannot be adapted to the context of divergence-based training.
>
> Nonetheless, please note that KL-, Renyi-, and Tsallis- divergences are also amenable to a degree of off-policy learning by implementing an importance sampling estimator — as long as the sampling policy can be directly evaluated at each individual trajectory. This is the case for, e.g., the widely-used $\epsilon$-greedy policy.
>
> Fundamentally, we see this as an instantiation of Wolpert’s No Free Lunch theorem: while, as we show, the minimization of statistical divergences is more sample-efficient and frequently leads to faster training convergence for GFlowNets, implementing these objectives constrains the user to the adoption of tractable off-policy sampling schemes.
>
> We will include this discussion in the revised manuscript.
>
> > The authors also assume that the backward policy is fixed (L153). While this is true in some cases, learning PB results in significant improvements to the learned sampler [3,4]. As far as I can tell, modifying the proposed objectives to accomodate learning the P_B is non-trivial.
>
> Thank you for the thought-provoking remark. In principle, learning $\log p_{B}$ is quite straightforward; the objective
>
> $$
> \min \mathcal{D}(p_{B}, p_{F})
> $$
>
> can be jointly minimized in both $p_{F}$ and $p_{B}$ for any divergence measure $\mathcal{D}$ in a VAE-style learning. However, implementing a learnable $p_{B}$ incurs a non-trivial computational overhead due to the additional number of backward passes for reverse-mode autodifferentiation. To illustrate this, we will run additional experiments for divergence-based training of GFlowNets with a learnable $p_{B}$ and report the results during the discussion period.
>
> > I appreciate the diversity of problems studied by the authors in their experiments, but there are some gaps in the empirical analysis. Specifically, the authors only include a TB baseline and not other objectives such as SubTB, DB which can perform better than TB (and have better training stability) in some cases. Additionally, the paper also does not include the VarGrad-style objective [5] which does away the need for estimating Z in TB.
>
> Thank you for your compliment. We have included DB, SubTB, and VarGrad as additional baselines for our experiments in Figures 1 and 2 of the attached PDF, in addition to two novel generative tasks.
>
> Remarkably, with the exception of the extremely sparse hypergrid environment, divergence-minimization algorithms lead to the fastest convergence rate among the tested learning objectives. We also note that, for the very sparse and hard-to-explore hypergrid, off-policy training is necessary and (as we discussed earlier) purely on-policy-based methods should be avoided. Nonetheless, for these problems, our empirical analysis shows that the mode-covering behavior of $\alpha$-divergences with large and negative $\alpha$ is very beneficial for speeding up training convergence (please refer to Figure 3 of the attached PDF). We will include these experiments in the revised manuscript.
>
> >  Moreover, the tasks consider relatively small tasks so it is not clear how scalable the proposed objectives are.
>
> Thanks for bringing up the discussion. We emphasize that the computation overhead incurred by the proposed gradient estimation techniques is negligible. In this sense, the divergence-based learning objectives are as scalable as their balance-based counterparts; please refer to Table 1 below. Results represent averages of 24 runs per criterion.
>
> Table 1: Runtime in minutes, avg over runs and environments.
> | criterion  | avg |
> |:-----------------|---------:|
> | KL         |9.5|
> | Renyi-$\alpha$   | 9.4|
> | Rev. KL          | 10.4|
> | Tsallis-$\alpha$ | 10.5|
> | TB               | 9.2|
>
>
> > L135 says TB requires estimating Z
>
> Indeed, the mathematical definition of TB and KL depends on the constant Z. Nevertheless, it is correct that Z is not needed in the context of gradient-based optimization of the KL. To see this, please note that we may write the reverse KL-divergence as
>
> $$
> \mathcal{D}\_{KL}[p\_{F} || p\_{B}] = \mathbb{E}\_{\tau \sim p\_{F}}[\log p\_{F}(\tau) - \log p\_{B}(\tau | x) R(x) / Z] = \mathbb{E}_{\tau \sim p\_{F}}[\log p\_{F}(\tau) - \log p\_{B}(\tau | x) R(x)] + \log Z.
> $$
>
> Consequently, when taking gradients with respect to the parameters of  $p_{F}$, the terms corresponding to $\log Z$ in the right-hand side of the equation above vanish, as it does not depend on $p_{F}$. In particular, $\log Z$ does not interfere with the problem of minimizing $\mathcal{D}\_{KL}[p_{F} || p\_{B}]$ and can be ignored. A similar argument holds for the forward KL-, Renyi-, and Tsallis divergences. We will update the revised manuscript to clarify this point.
>
> > L146: [1] would be a more appropriate reference here I think?; L157: Missing reference to [2]
>
> We agree! These important references will be incorporated into the revised manuscript.
>
> > What is the computational performance (in terms of runtime) of the proposed objectives relative to TB?
>
> Please refer to Table 1 above. As we remarked earlier, our approach adds a negligible computational overhead to the training of GFlowNets.

---

> > ### Comment · Reviewer_3sBX · 2024-08-08
> > **Response to rebuttal**
> >
> > Thanks for the comments. I have a few follow-ups
> >
> > > amenable to a degree of off-policy learning by implementing an importance sampling estimator
> >
> > Indeed one can always use importance sampling to do off-policy training with an on-policy objective (e.g. as noted in [1]) but IS also introduces other challenges (e.g. high variance). So as you note there is certainly a trade-off. However recent work has also illustrated the importance of off-policy samples for training GFlowNets in challenging domains [2,3] which might make the objectives like TB better suited.
> >
> > >  we will run additional experiments for divergence-based training of GFlowNets with a learnable $p_B$
> >
> > Looking forward to the results.
> >
> > > computation overhead incurred by the proposed gradient estimation techniques is negligible
> >
> > The compute overhead is indeed not high, but my comment was more about the scale of the problems used in the paper. Prior work on learning objectives has considered much larger and challenging problems. I see the authors have added the hypergrid and causal DAG tasks but do not provide details about the size of the problems? What size was the hypergrid and the number of variables considered for the causal DAG task?
> >
> > > mathematical definition of TB and KL depends on the constant Z
> >
> > I agree that the Z does not play a role in the optimization but I would note that in a similar fashion TB can also be optimized without the Z (i.e. VarGrad)
> >
> > In the common response the authors claim "broadest experimental evaluation of GFlowNet objectives in the literature" but I think this loses a lot of nuance. While the paper does consider 6 tasks, they are all much smaller (and potentially simpler) than tasks considered in prior work (e.g. much larger molecular optimization and sequence design). This nuance is critical and I hope the authors avoid making sweeping claims.
> >
> > [1] GFlowNets and variational inference, ICLR 2023.
> >
> > [2] Amortizing intractable inference in large language models, ICLR 2024.
> >
> > [3] Improved off-policy training of diffusion samplers, arXiv:2402.05098

---

> ### Author Response · Authors · 2024-08-10
>
> Thank you for engaging in the discussion!
>
> > However recent work has also illustrated the importance of off-policy samples for training GFlowNets in challenging domains [2,3] which might make the objectives like TB better suited.
>
> We agree! We will include our discussion regarding the trade-off between off- and on-policy learning in the revised manuscript.
>
> > Looking forward to the results.
>
> Below, we report the results comparing the accuracy of the GFlowNet when $\log p_{B}$ is either learned or fixed, respectively, for the tasks of set generation and structure learning (Tables 2 and 3). We observed related results for the task of BPI - with similar performances for learnable and uniform $p_{B}$’s - and we will include these experiments in the revised manuscript. Although they show that jointly minimizing the learning objective wrt both $\log p_{B}$ and $\log p_{F}$ is a sound strategy for divergence-based measures, they do not reveal clear benefits favoring its implementation.
>
> Table 2: Set generation. Results are averaged across 3 runs.
> | | learn | unif |
> |---|---|---|
> | TB | 0.29 ± 0.02 | 0.13 ± 0.00 |
> | Reverse KL | 0.09 ± 0.01 | 0.03 ± 0.00 |
> | Renyi-0.5 | 0.09 ± 0.01 | 0.03 ± 0.00 |
>
> Table 3: DAGs. Results are averaged across 3 runs.
> | | learn | unif |
> |---|---|---|
> | TB | 0.43 ± 0.11 | 0.47 ± 0.13 |
> | Reverse KL | 0.31 ± 0.14 | 0.32 ± 0.14 |
> | Renyi-0.5 | 0.14 ± 0.02 | 0.14 ± 0.02 |
>
> To further investigate the shape of the learned backward policy, we computed the expected entropy of $p_{B}(\tau | x)$ under the learned marginal $p_{T}$ over terminal states, i.e., $\mathbb{E}\_{x \sim p\_{T}}[\mathbb{E}\_{\tau \sim p\_{B}(\cdot | x)}[ - \log p\_{B}(\tau | x)]]$. Intuitively, a highly entropic policy is closer to a uniform policy, following the analysis of Shen et al. [1]. In this sense, the results in Tables 4 and 5 below suggest that the learned backward policy closely resembles an uniform distribution.
>
>
> Nonetheless, we believe that a deeper investigation of the potential advantages of learning $\log p_{B}$ is a relevant and interesting research direction.
>
> Table 4: Set generation
> | | learn | unif |
> |---|---|---|
> | TB | 30.61 ± 0.01 | 30.67 ± 0.00 |
> | Reverse KL | 30.61 ± 0.01 | 30.67 ± 0.00 |
> | Renyi | 30.61 ± 0.01 | 30.67 ± 0.00 |
>
> Table 5: DAGs
> | | learn | unif |
> |---|---|---|
> | TB | 18.53 ± 3.61 | 18.55 ± 3.61 |
> | Reverse KL | 19.26 ± 3.52 | 19.27 ± 3.52 |
> | Renyi | 19.29 ± 3.42 | 19.31 ± 3.43 |
>
> [1] Towards Understanding and Improving GFlowNet Training. Shen et al. ICML 2023
>
> > The compute overhead is indeed not high, but my comment was more about the scale of the problems used in the paper.
>
> Apologies for the oversight. For the DAG task, we considered graphs with 6 nodes; the target distribution’s support contains approximately 3.5M graphs. For the hypergrid task, we considered a 12 x 12 grid in Figure 1 and a 9 x 9 grid in Figure 3 of the rebuttal PDF; similarly to [2, 3], we set $R_{o} = 10^{-3}$. We will include these details in the revised manuscript.
>
> [2] GFlowNets and Variational Inference. Malkin et al. ICLR 2023.
>
> [3] Trajectory balance: Improved Credit Assignment in GFlowNets. Malkin et al. NeurIPS 2022.
>
> >  This nuance is critical and I hope the authors avoid making sweeping claims.
>
> Thanks for the advice. We only meant to flesh out the diversity of tasks and baselines of our experimental campaign in the rebuttal.
>
> We are very grateful for your detailed feedback and contribution towards strengthening our paper!

---

> ### Comment · Reviewer_3sBX · 2024-08-13
> **Response**
>
> Sorry for the delay in my response, but I appreciate the detailed reply.
>
> > learned $p_B$
>
> Thanks for sharing these results. I am a bit confused that on the set generation experiments for TB uniform does better than learned $p_B$. But indeed these results are interesting and could be an interesting avenue for future work.
>
> > For the DAG task, we considered graphs with 6 nodes.
>
> I appreciate the authors including these two tasks but I should emphasize that as with all the other experiments in the paper, these tasks are significantly smaller (and thus potentially easier) than prior work. For instance, in the TB paper the hypergrids considered were $8\times 8\times 8\times 8$ and $64\times 64$, not to mention much larger sequence design and molecular design tasks. I stand by my original review that the major weakness of the paper is the scale and difficulty of tasks considered in the empirical evaluation.

---

> ### Author Response · Authors · 2024-08-13
>
> Thank you for your continued engagement.
>
> While we acknowledge many studies consider problems on a larger scale, we note that our empirical analysis primarily focuses on assessing GFlowNet’s distributional accuracy during training.
>
> For problems of much larger scale, such as molecule generation (with approximately $10^{16}$ terminal states), it is not feasible to accurately measure the goodness-of-fit of a trained GFlowNet. This is because it would require exhaustively enumerating the target’s support, which is necessary to assess the total variation distance between the learned and target distributions.
> Therefore, we have constrained our analysis to the environments discussed in the rebuttal PDF, which we believe are sufficiently diverse.
>
> We will discuss this matter at the end of the revised manuscript. Thank you very much for your detailed and constructive feedback.

---

### Official Review · Reviewer_HRNm · 2024-07-13

**Soundness:** 3
**Presentation:** 4
**Contribution:** 4
**Rating:** 7
**Confidence:** 4

**Summary:**

This paper investigates the potential of using a variety of divergence measures directly as training losses for GFlowNets, relying on many of the connections made between GFlowNets and variational inference. Training GFlowNets essentially consists in enforcing balance/flow-matching conditions between a proposal and a target distribution given some data. Given the links between GFlowNets and variational inference, training the GFlowNet to directly minimize the KL divergence was observed to result in biased and high-variance estimators in general. This paper aims to verify the latter claim, for different divergence measures (KL, reverse KL, Tsallis-\$\alpha\$, Renyi-\$\alpha\$). The paper alleviates the afore-mentioned limits by proposing low-variance gradient estimates to that of the latter quantities through control variates. Finally, the paper verifies the proposed methods experimentally through an extensive set of experiments.

**Strengths:**

- The paper widens the scope of the theoretical links between GFlowNets and variational inference beyond the assumption of finitely supported measures.
- The authors alleviate the high variance of divergence measures' gradients in practice, relying on control variates.
- The paper is very well written and presented and is easy to follow.
- The experimental setup is exhaustive, and shows the effect of each of the proposed design choices.

**Weaknesses:**

See questions.

**Questions:**

- How is the diversity of samples impacted?
- There should be experiments that show that the different behaviors of each method with respected to the number of discovered modes throughout training (that should be compared to TB too!). For instance, similarly to Figure 4 in (Malkin et al., https://arxiv.org/pdf/2201.13259).

**Limitations:**

Limitations adequately addressed throughout the paper.

---

> ### Author Rebuttal · Authors · 2024-08-07
>
> Thank you for valuable suggestion and review. We did our best to address each of your questions, and extended our experiments following your suggestion. Please let us know if you have other questions or require further clarification.
>
> > How is the diversity of samples impacted?
>
> Considering that the *diversity of the samples* could be approached from different perspectives, we made our best effort to address your question in a broad sense.
>
> **Support coverage: divergence measures versus TB.** Overall, we observe that our divergence-based learning procedures yield better support coverage than the TB, consequently capturing shapes more accurately. Figure 4 in the manuscript illustrates this phenomenon for a banana-shaped target distribution.
>
> **Number of visited modes.** Compared to balance-based objectives, we observe that our divergence-based training leads to faster increase in NoM and average reward of the top-K samples during the training process in the majority of environments — Figure 2 (rebuttal PDF). This metrics can be interpreted as indicative of diversity of the high-reward samples.
>
> **Impact of $\alpha$ on sample diversity.** In general, $\alpha$ allows us to modulate between mode-seeking and mass-covering behaviors, directly impacting the sample diversity. Low $\alpha$ leads to capturing more modes, while exceedingly high $\alpha$ may cause mode collapse — see, e.g., Figure 1 (manuscript) and Figure 3 (rebuttal PDF).
>
> > There should be experiments that show that the different behaviors of each method with respected to the number of discovered modes throughout training (that should be compared to TB too!). For instance, similarly to Figure 4 in (Malkin et al., https://arxiv.org/pdf/2201.13259).
>
> Thank you for the question and suggestion. We have included an analysis of the number of modes (NoM) discovered as a function of time in Figure 2 (rebuttal PDF). Overall, except for the Hypergrid environment, the divergence measures (Rev. KL, KL, Renyi-α and Tsallis-α, w/ α=0.5) have a higher NoM visited earlier in the training process than the balance-based losses (TB, DB, SubTB, VarGrad). Furthermore, their NoM learning curves display a faster rate of increase in the Sets and Sequences environment. For completeness, we include the average reward for the K highest scoring samples (top-K), showing similar results as the NoM.
>
> The distinct behavior for Renyi-α and Tsallis-α in the hypergrid environment could be explained by the results in Figure 3 (rebuttal PDF), since α=0.5 leads to mode collapse.

---

### Official Review · Reviewer_NozV · 2024-07-16

**Soundness:** 2
**Presentation:** 2
**Contribution:** 2
**Rating:** 4
**Confidence:** 5

**Summary:**

This paper investigates divergence measures as learning objectives for Generative Flow Networks, which are amortized inference models designed for sampling from unnormalized distributions over composable objects. The authors review four divergence measures - Renyi-\alpha, Tsallis-\alpha, reverse and forward Kullback-Leibler divergences, and design estimators for their stochastic gradients in the context of training enerative Flow Networks. The authors verify that minimizing these divergences yields correct and empirically effective training schemes on several toy environment, and show that it often lead to faster convergence than previously proposed optimization methods. The authors also design control variates based on REINFORCE and score-matching estimators to reduce gradient variance.

**Strengths:**

- The paper provides evaluation of different divergence measures as learning objectives for GFlowNets, showcasing their effectiveness in improving training convergence in several toy environments.

- The authors develop control variates for reducing the variance of gradient estimates.

**Weaknesses:**

1. While the paper provides empirical evidence for the effectiveness of divergence-based objectives, a more extensive comparison with traditional GFlowNet training methods across a wider range of datasets and applications would strengthen the claims.

2. The choice of the \alpha parameter in Renyi-\alpha and Tsallis-\alpha divergences is not extensively explored. More insights into the impact of \alpha on the learning dynamics and guidance on selecting an appropriate value would be beneficial.

3. The computational overhead introduced by the control variates and their impact on training time is not explicitly discussed.

**Questions:**

1. How do the proposed divergence-based objectives perform on more complex and high-dimensional datasets commonly used in fields such as drug discovery and natural language processing?

2. Can the control variate techniques be extended to other GFlowNet training objectives beyond the divergence measures considered in this paper?

3. How does the choice of the \alpha parameter affect the learned GFlowNet's ability to capture multi-modal target distributions or discover diverse high-quality samples in combinatorial optimization tasks?

4. Are there any theoretical guarantees or bounds on the sample complexity or convergence rates when using the proposed divergence-based objectives for training GFlowNets?

**Limitations:**

Yes.

---

> ### Author Rebuttal · Authors · 2024-08-07
>
> Thank you for your feedback. We hope our answers address your concerns and elevate your appraisal of our work. Otherwise, we would be happy to engage further.
>
> > … a more extensive comparison across a wider range of datasets and applications would strengthen the claims.
>
> We have included SubTB [1], DB [2], and VarGrad [3,4] losses as baselines and included the hypergrid [5] and causal DAG environments [6]. With this, our experimental campaign is broader than most works in the GFlowNet literature, including six different environments with continuous and discrete target distributions. In contrast, most works in the GFlowNet literature use three to four environments [1, 2, 4, 5, 6, 7, 8], typically using discrete targets and comparing against a small set of learning objectives.
>
> Given our comprehensive experimental suite, we can confidently conclude that divergence-based objectives are beneficial for training GFlowNets as long as they are equipped with appropriate control variates, contrary to the belief established by the results in [18].
>
> > on choosing $\alpha$ and its effect on GFlowNet's ability to capture modes
>
> Thank you for the opportunity to improve our discussion on the role of $\alpha$. To the best of our knowledge, delineating specific guidelines for choosing optimal $\alpha$ for VI is still an open problem [11, 12]. Our submission (Lines 164-186 and Figure 1) discusses how $\alpha$ modulates mode-seeking vs. mass-covering behavior. This is a necessary starting point practitioners considering the choice of $\alpha$.
>
> To further illustrate the effect of varying $\alpha$, we executed a series of experiments for the hypergrid task – please refer to Fig 3 of the rebuttal PDF. Overall, our analyses indicate that, for sparse distributions, large negative values of $\alpha$ perform better – on par with the mass-covering effect of the corresponding divergence measure. In addition, we also ran experiments for the set generation, sequence design, and BPI tasks with varying values of $\alpha$, observing similar results.
>
> We will include all additional results in the revised manuscript.
>
> > computational cost of control variates
>
> In practice, the computational overhead of our variance reduction techniques is negligible. We report in Table 1 the runtime (in minutes) of the training process averaged over all environments considered in the experiments, a total of 120 runs, for each learning objective. The avg runtime confirms the small overhead of our variance reduction method. Furthermore, the control variate significantly speeds up the convergence, as shown in Fig. 5 of our manuscript (Section 5.3).
>
> Table 1: Runtime in minutes, avg over runs and environments.
> | criterion        |     avg |
> |:-----------------|---------:|
> | KL               | 9.5|
> | Renyi-$\alpha$   | 9.4|
> | Rev. KL          | 10.4|
> | Tsallis-$\alpha$ | 10.5|
> | TB               | 9.2|
>
> > Performance in complex and high-dimensional tasks
>
> Thanks for the question. We would like to clarify that our adopted environments represent both realistic and prototypical for GFlowNets. For example, the phylogenetic tree inference [7] and sequence generation [8], presented in Figure 3, are real-world tasks with papers assessing the effectiveness of GFlowNets in solving them [7, 8]. Also, we estimate  ~$10^{7}$  possible final objects in the set generation and sequence design tasks, highlighting their realistic scales.
>
> We have now included hypergrid [1] and Causal DAG [2] as novel tasks; and DB, SubTB, and VarGrad as baselines. Strikingly, Renyi, Tsallis, and KL perform on par with or better than balance-based losses in most scenarios (Figs 1 and 2, rebuttal PDF).
>
> > Can the control variate techniques be extended to [balance-based] GFlowNet training objectives [...]?
>
> In principle, yes. However, it is unclear how to devise efficient CVs for balance-based objectives.
>
> Firstly, conventional objectives rely on off-policy sampling, and it is mostly unclear how to define a control variate that (i) has zero expectation under a chosen policy and (ii) is correlated to the objective's gradient. Secondly, as shown in Fig 6 (supplement), the smoothness of the learning curve for TB suggests that gradient variance is not an issue for balance-based objectives. Notably, it is well-known that the high variance for divergence-based objective stems from the score-function estimator, $\mathbb{E}\_{\tau \sim p_{F}}[\nabla \log p_{F}(\tau)]$, which is not present in the gradient of balance-based objectives such as TB.
>
> > Theoretical guarantees on the training convergence
>
> Developing convergence rate analysis for GFlowNets remains an open problem. Many works focus on designing sample-efficient learning objectives for GFlowNets, but none provide convergence rate analyses.
>
> Also, establishing convergence rates for VI for generic distribution classes is an open problem. Initial works [17] assumed strong convexity in a lower-bound problem relevant to GANs, obtaining geometric convergence rates. More recently, [16] obtained guarantees for BBVI and Gaussian approximations. Adapting these results to GFlowNet is a fruitful line of work, meriting an investigation of its own.
>
> [1] Learning GFlowNets from partial episodes for improved convergence and stability. ICML 2023
>
> [2] GFlowNet Foundations. JMLR 2023
>
> [3] VarGrad: A Low-Variance Gradient Estimator for VI. NeurIPS 2020
>
> [4] Robust Scheduling with GFlowNets. ICLR 2023
>
> [5] TB: Improved credit assignment in GFlowNets. NeurIPS 2022
>
> [6] Bayesian structure learning with GFlowNets. UAI 2022
>
> [7] PhyloGFN: Phylogenetic inference with GFlowNets. ICLR 2024
>
> [5] Biological sequence design with GFlowNets. ICML 2022
>
> [12] Rényi Divergence VI. In NeurIPS 2016
>
> [13] Meta-learning divergences for VI. In AISTATS 2021
>
> [16] Provable convergence guarantees for BBVI. NeurIPS 2024
>
> [17] g-GAN: Training generative neural samplers using variational divergence minimization. NeurIPS 2016
>
> [18] GFlowNets and VI.  ICLR 2023

---

### Author Rebuttal · Authors · 2024-08-07

Dear reviewers and AC,

We appreciate that reviewers evaluation of our work as both theoretically principled [Gn1e, HRNm]  and empirically well-grounded [Gn1e, 3sBX, NozV], expanding the link between VI and GFlowNets [Gn1e, 3sBX, HRNm], with practical contribution of effective control variates for variance reduction [Gn1e, 3sBX, HRNm, NozV], presented with clarity [3sBX, HRNm], and with reproducible results [3sBX] supported by a comprehensive range of experiments [Gn1e, 3sBX, HRNm].

We present a summary of the central points in the discussion and how they were addressed.

1. Reviewers NozV, 3sBx, Gn1e, and HRNm suggested including additional benchmark tasks, baseline objectives, and evaluation metrics.

    a. Firstly, we included the hypergrid and DAG environments in our experimental suite.

    b. Secondly, we added SubTB, VarGrad, and DB among our baselines.

    c. Thirdly, we measured the number of modes and the average reward of the highest $K$ scoring samples during training (top-$K$).

    d. Our results confirm that, for the majority of problems, divergence-based objectives lead to faster training convergence than balance-based ones. To the best of our knowledge, this is the **broadest experimental evaluation of GFlowNet objectives in the literature**.

2. Reviewers NozV and 3sBx requested a runtime analysis to quantify the computational overhead introduced by the control variates. We provide a table in their respective answers showing that their overhead is negligible for both the KL- and $\alpha$-divergences.

3. Reviewers NozV and Gn1e asked for an assessment of the effect of $\alpha$ on training convergence and guidelines for choosing it. Our novel experiments ilustrate how varying $\alpha$ modulates the balance between mode-seeking and mass-covering behaviors and that large negative $\alpha$ values are more suited to sparse target distributions (Fig. 1, main text; Fig. 3, rebuttal PDF).

4. Reviewers NozV and Gn1e questioned the comparative benefit in real-world and high-dimensional tasks; reviewer 3sBX raised concerns about scalability. Beyond the extensive experiments and realistic tasks such as BPI, we included the hypergrid and Causal DAG as additional environments. Compared to baselines (TB, DB, SubTB, and VarGrad), divergence-minimization algorithms lead to faster training convergence in most problems (Figs 1 and 2, rebuttal PDF).

We would also like to thank the reviewers again for their invaluable feedback and their help in strengthening our work.

---

### Comment · Area_Chair_bEWv · 2024-08-08
**Author-Reviewer Discussion Phase (ends Aug 13 midnight AOE)**

Dear reviewers,

Thank you for your efforts so far! The authors have performed a number of additional experimental evaluations in response to your reviews. Could you follow up with them to ask any clarifying questions that may further inform your opinion on the paper?

With thanks,
AC

---

### Author Response · Authors · 2024-08-14

Dear reviewers and conference chairs,

We would like to thank you for your service and dedication to the peer-reviewing process. We understand that some reviewers could not go through our rebuttal during the discussion period. Nonetheless, we made our best effort to incorporate all suggestions and believe the reviewers' feedback was paramount to strengthening our work and clarifying its importance.

Best regards,

The authors

---

### Decision · Program_Chairs · 2024-09-25

**Decision:**

Accept (poster)

**Comment:**

This paper f-divergence based objectives for GFlowNets (GFNs). These provide an alternative to the standard trajectory-balance objective for GFN and define generalizations of variational objectives for GFNs based on KL divergences. The authors consider Renyi-$\alpha$, Tsalli-$\alpha$, as well as the forward and reverse KL. The authors derive gradients for these objectives in the case of a fixed backward policy and on-policy samples, and develop control variates that reduce the variance of gradient estimates. Experiments demonstrate that using these objectives can accelerate convergence and enhance stability of training.

Reviewers did not fully arrive at consensus regarding this submission. Overall reviewers were sympathetic to the exploration of alternative objectives for GFNs presented in the paper, as well the contributions in terms of developing gradient estimators and control variables. However, but reviewers also noted limitations of the approach in terms of the setting with fixed backward policy and on-policy samples, as well the experimental evaluation, which primarily focuses on relatively simple toy problems. Overall two reviewers (HRNm and 3sBX) support acceptance, with 3sBX noting during the AC discussion phase that in spite of shortcomings, they consider the paper to make enough contributions to warrant acceptance. Out of the more critical reviewers, Gn1e raised their score after the author response. Reviewer NozV, who was the most critical of the submission in their review, did not engage with the author response or the AC. As far as the AC can determine, at least some of their concerns were addressed directly by the author response, and as such the AC will somewhat discount the score from this reviewer.

The AC took a quick look at the submission, which appears to be well-presented piece of work that makes a reasonable set of contributions to this space. This seems above the bar for acceptance. Do please incorporate the revisions that were discussed in the author response.